# GTP-dependent formation of straight tubulin oligomers leads to microtubule nucleation

Rie Ayukawa[1]*, Seigo Iwata[1]*, Hiroshi Imai[2]*, Shinji Kamimura[2], Masahito Hayashi[1], Kien Xuan Ngo[1], Itsushi Minoura[1], Seiichi Uchimura[1], Tsukasa Makino[1], Mikako Shirouzu[3], Hideki Shigematsu[3], Ken Sekimoto[4,5], Benoît Gigant[6], and Etsuko Muto[1]

Nucleation of microtubules (MTs) is essential for cellular activities, but its mechanism is unknown because of the difficulty involved in capturing rare stochastic events in the early stage of polymerization. Here, combining rapid flush negative stain electron microscopy (EM) and kinetic analysis, we demonstrate that the formation of straight oligomers of critical size is essential for nucleation. Both GDP and GTP tubulin form single-stranded oligomers with a broad range of curvatures, but upon nucleation, the curvature distribution of GTP oligomers is shifted to produce a minor population of straight oligomers. With tubulin having the Y222F mutation in the β subunit, the proportion of straight oligomers increases and nucleation accelerates. Our results support a model in which GTP binding generates a minor population of straight oligomers compatible with lateral association and further growth to MTs. This study suggests that cellular factors involved in nucleation promote it via stabilization of straight oligomers.

## Introduction

In eukaryotes, dynamic arrangement of microtubule (MT) organization in proper timing and location is critical for various cellular functions, such as cell shape determination and chromosome segregation. The high degree of plasticity of the MT network relies on the nucleation and polymerization/depolymerization of individual MT filaments. When the regulation of MT dynamics is disrupted, it causes pathological disorders such as cancer (Schukken et al., 2020) and neurodegenerative diseases (Kounakis and Tavernarakis, 2019). Therefore, understanding the molecular mechanism of MT dynamics is important for developing an effective treatment of these diseases.

While the catastrophic transition to depolymerization has been well characterized as "dynamic instability" (Brouhard, 2015; Mitchison and Kirschner, 1984), the inverse process involving the nucleation of MTs from tubulin is still poorly understood. Nucleation in general is a stochastic process in which the thermodynamically unfavorable growth of precritical molecular composites turns into favorable growth via the formation of critical nuclei (Burton, 1977; Langer, 1969). In this paper, we solve a long-standing important question: How does GTP tubulin nucleate MTs in vitro? Despite its central importance, the question remained unresolved for decades because the nucleation intermediates around the critical nuclei exist only transiently, so capturing them is a major challenge.

Compared with the rapidly growing interest among cell biologists in the mechanism of in vivo nucleation mediated by the template of the γ-tubulin ring complex (γTuRC) and other cellular factors (Brunet et al., 2004; Flor-Parra et al., 2018; Kollman et al., 2011; Roostalu et al., 2015; Schatz et al., 2003; Thawani et al., 2018; Zheng et al., 1995), the interest in the mechanism of spontaneous nucleation in vitro has been scarce. However, the kinetics of the templated assembly show a kind of time lag between the onset of the reaction and the start of MT growth from the template (Wieczorek et al., 2015; Woodruff et al., 2017), suggesting that, even in the presence of a template, the initial phase of assembly is thermodynamically unfavorable until the nucleation intermediate reaches a critical size. In other words, nucleation in cells and nucleation in vitro should have a fundamental scheme in common (Roostalu and Surrey, 2017). As a first step toward the understanding of the mechanism of

........................................................................................................................................................................................................................................
[1]Laboratory for Molecular Biophysics, RIKEN Center for Brain Science, Saitama, Japan;   [2]Department of Biological Sciences, Faculty of Science and Engineering, Chuo University, Tokyo, Japan;   [3]Laboratory for Protein Functional and Structural Biology, RIKEN Center for Biosystems Dynamics Research, Yokohama, Japan;   [4]Matière et Systèmes Complexes (MSC), CNRS UMR 7057, Université de Paris, Paris, France;   [5]Gulliver, CNRS UMR 7083, ESPCI Paris and Université Paris Sciences et Lettres, Paris, France;   [6]Université Paris-Saclay, CEA, CNRS, Institute for Integrative Biology of the Cell (I2BC), Gif-sur-Yvette, France.

*R. Ayukawa, S. Iwata, and H. Imai contributed equally to this paper;   Correspondence to Etsuko Muto: etsuko.muto@riken.jp;   Benoît Gigant: benoit.gigant@i2bc.paris-saclay.fr;   Ken Sekimoto: ken.sekimoto@espci.fr;   H. Imai's present address is Department of Biological Sciences, Graduate School of Science, Osaka University, Osaka, Japan; M. Hayashi's present address is Department of Frontier Bioscience, Hosei University, Tokyo, Japan; K.X. Ngo's present address is WPI Nano Life Science Institute, Kanazawa University, Kanazawa, Japan; I. Minoura's present address is Product Development Division, Goryo Chemical, Inc., Sapporo, Japan; S. Uchimura's present address is CPI Company, DAICEL Corporation, Hyogo, Japan;   T. Makino's present address is Department of Cell Biology and Anatomy, Graduate School of Medicine, University of Tokyo, Tokyo, Japan;   H. Shigematsu's present address is Life Science Research Infrastructure Group, RIKEN SPring-8 Center, Hyogo, Japan.

**Rockefeller University Press**
J. Cell Biol. 2021 Vol. 220 No. 4   e202007033



**https://doi.org/10.1083/jcb.202007033**   1 of 19

nucleation in vivo, we clarify the basic scheme of nucleation using a simple in vitro system composed only of tubulin.

GTP tubulin is known to assume a curved conformation in solution (Manuel Andreu et al., 1989; Rice et al., 2008) and a straight conformation when integrated in the MT lattice (Nogales et al., 1999; Wang and Nogales, 2005). Therefore, during some step in polymerization, GTP tubulin is expected to change its conformation. However, when and how straightening occurs have been a subject of long debate (Bennett et al., 2009; Brouhard and Rice, 2014; Rice et al., 2008). In this work, we characterized the oligomers, and in particular their curvature, that form before MT assembly. Because the nucleation intermediates that indeed grow to MTs appear only at high tubulin concentrations (~10 µM), their visualization is impossible by ordinary imaging techniques, which require much less concentrated samples. We overcome this difficulty by using rapid flush negative stain EM (Frado and Craig, 1992), where we quickly dilute the sample to the concentration optimal for EM observation (approximately submicromolar).

Our analysis shows that a GTP-dependent curvature shift occurs in single-stranded oligomers in the very early stage of nucleation. Both GTP and GDP tubulin assemble into single-stranded oligomers of various lengths and curvatures, but it is only with GTP tubulin that a minor population of straight oligomers appears. The proportion of straight oligomers was increased by the Y222F mutation in β-tubulin (sequential numbering, with Y222 corresponding to Y224 in the numbering used in Nawrotek et al. 2011), which paralleled the acceleration of nucleation by this mutation. These results collectively indicate that among dimers and oligomers with variable curvatures, only the rare straight oligomers can overcome the energy barrier for nucleation. Preceding our study, earlier works using negative stain EM suggested the involvement of short oligomers in nucleation (Mozziconacci et al., 2008; Portran et al., 2017), and kinetic analysis gave an estimate for the size of critical nucleus (Voter and Erickson, 1984), but it was not clear whether and how the oligomers in the electron micrograph relate to the critical nuclei. By linking statistical analyses of oligomers and the kinetic analyses, we succeeded in deciphering how straight oligomers reaching a critical size can become a platform for thermodynamically favorable growth. That mechanism also likely underlies the nucleation in vivo mediated by various cellular factors.

## Results

### Structure and kinetics of WT tubulin and of the Y222F mutant

It has been proposed that GTP binding promotes MT assembly by fostering a conformational switch of the β-tubulin T5 loop, a loop involved in tubulin–tubulin longitudinal contacts (Fig. 1, A and B; Nawrotek et al., 2011). The crystal structure of GDP tubulin indicates that the T5 loop fluctuates between two conformations ("in" and "out"; Fig. 1 A). Upon GTP binding, the interaction of residue D177 in the T5 loop with residue Y222 from the H7 helix is broken, and thus the "out" conformation is favored. With the T5 loop in "out" conformation, the negatively charged residue D177 in β-tubulin is exposed toward the solvent,

likely mediating the incoming tubulin dimer having positive charges on the α-tubulin interface (in particular, from K336 and K352) to establish a longitudinal contact (Natarajan et al., 2013; Nawrotek et al., 2011; Fig. 1 B). Based on this assumption, we expected that the residue substitution Y222F may produce a stronger bias toward the "out" conformation in the T5 loop, leading to rapid MT nucleation.

To test this idea, we took advantage of the recent development of a method for the expression and purification of recombinant tubulin (Minoura et al., 2013; Minoura et al., 2016). Here, we modified our original method for the preparation of human α1β3 tubulin to purify *Drosophila* α1β1 tubulin. *Drosophila* tubulin has several technical advantages over human tubulin: its yield from host insect cells is better, and it can polymerize at room temperature. Both *Drosophila* WT and Y222F mutant tubulins were produced and purified (Fig. S1).

We first determined the crystal structure of *Drosophila* WT tubulin within $T_2R$, a complex composed of two tubulin heterodimers and one stathmin-like domain of the RB3 protein. The structure showed the nucleotide-dependent movement of the T5 loop in the β subunit exposed to solvent in $T_2R$ (β2 in Fig. 1 C); the loop was in the "out" and the "in" conformations in the presence of the stable GTP analogue guanylyl-(α, β)-methylene-diphosphonate (GMPCPP) and in the GDP state, respectively (Fig. 1, C–F; and Table S1). This is similar to the nucleotide-dependent structural change of the T5 loop observed for mammalian brain tubulin (Nawrotek et al., 2011), though the structural difference between the two states is more pronounced in *Drosophila* tubulin. In the case of the Y222F mutant, the T5 loop was "out" with bound GTP (Fig. 1 H; note that we did not succeed in obtaining crystals of Y222F with bound GMPCPP; see Materials and methods for details), but was poorly defined in the electron density maps in the GDP state (Fig. 1 G). Therefore, the Y222F mutation destabilized the T5 loop "in" conformation, but in the absence of GTP, it did not force this loop into the "out" conformation. In short, the Y222F mutation gave a bias in the structural equilibrium of the T5 loop toward the "out" conformation, as we aimed for.

We next compared the time course of the assembly of WT and Y222F tubulins by monitoring the turbidity of the tubulin solution (Fig. 1, I and J). Because Y222F GTP tubulin readily polymerized as soon as we transferred the tubulin solution from 4°C to 25°C, we had to monitor the assembly at concentrations significantly lower than the concentrations used for WT tubulin (2–5 µM and 10–30 µM for Y222F and WT tubulin, respectively). Despite the lower concentration range employed, Y222F tubulin assembled much faster than WT tubulin. For both proteins, the turbidity plateau was a linear function of the initial tubulin concentration, yielding critical concentrations (x intercept) of 4.7 ± 0.5 and 0.19 ± 0.15 µM (mean ± SD) for WT and Y222F tubulin, respectively (Fig. 1 K).

The rate of nucleation was assessed from the inverse of the time required for the turbidity to reach 10% of the plateau value ($1/T_{10\%}$; Voter and Erickson, 1984), which was log–log plotted as a function of the initial tubulin concentration (Fig. 1 L). For Y222F tubulin, $1/T_{10\%}$ increased in the lower concentration range and showed higher sensitivity to tubulin concentration

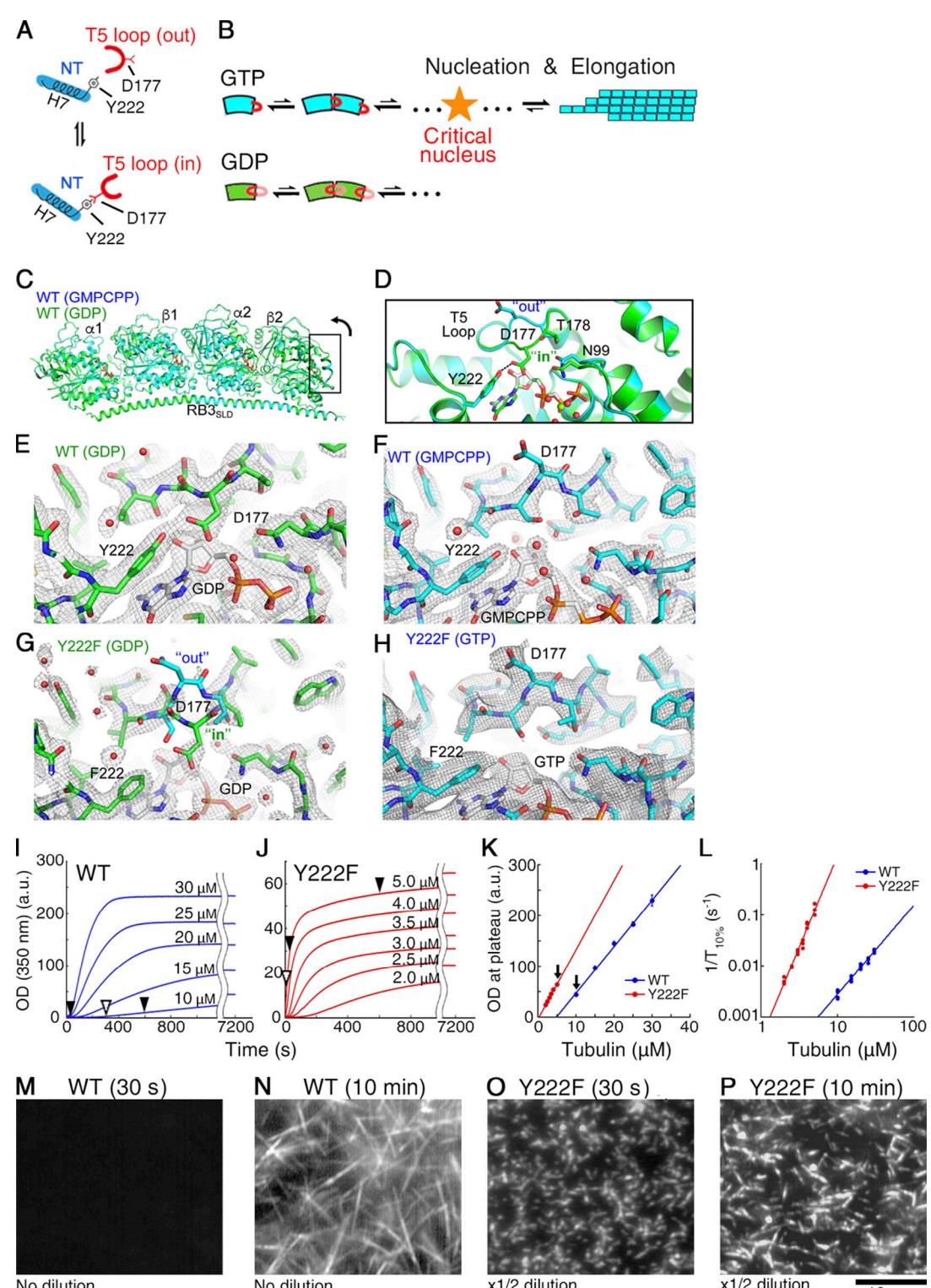

Figure 1. **The Y222F mutation modulates the structure of the T5 loop and accelerates nucleation. (A)** Schematic picture of the nucleotide-dependent regulation of the T5 loop in β-tubulin (Nawrotek et al., 2011). The "in"/"out" conformation of the T5 loop is regulated by formation/dissociation, respectively, of a hydrogen bond between the D177 residue in the T5 loop and the Y222 residue in H7. GTP/GDP binding biases the structural equilibrium of the T5 loop toward the "out"/"in" conformation, respectively. NT, nucleotide. **(B)** Schematic picture of the nucleotide-dependent regulation of the pathways of oligomer assembly. Only for GTP tubulin with the T5 loop in the "out" conformation does growth become thermodynamically favorable, as the oligomer exceeds the critical size. **(C–H)** *Drosophila* WT (C–F) and Y222F (G and H) tubulin structures within the $T_2R$ complex. **(C and D)** Superposition of the GMPCPP and GDP tubulin structures (C, overview; D, close-up of the part framed in C rotated 90°). **(E–H)** Two $F_{obs}$-$F_{calc}$ electron density maps contoured at the 1 σ level and centered on the β2 nucleotide-binding site and T5 loop. In the case of Y222F(GDP) tubulin (G), the T5 loop residue D177 and neighboring residues are not defined in the electron density map, suggesting that this loop is mobile. Both the "in" and "out" conformations of the T5 loop are drawn as a reference. **(I and J)** Time course of the

polymerization of WT (I) and Y222F tubulin (J) monitored by OD₃₅₀. At times indicated by black arrowheads, the MTs were checked under the darkfield microscope (M–P) and by negative stain EM (Fig. 2, A–C). At times indicated by white arrowheads, the oligomers were imaged using negative stain EM (Fig. 2, E–G; Fig. 4 D; and Fig. 6, A–J). **(K)** Plateau value of turbidity (I and J) plotted as a function of the initial tubulin concentration, showing the critical concentration (x intercept) of WT and Y222F tubulin to be 4.7 ± 0.5 and 0.19 ± 0.15 (μM), respectively. Error bars represent SD ($n$ = 3–5 for each concentration). The arrows indicate the pair of experiments compared in Fig. 2, E–G. **(L)** Log–log plot of the inverse of the time required for the turbidity to reach 10% of its plateau value (1/$T_{10\%}$) vs. initial tubulin concentration (Voter and Erickson, 1984). **(M–P)** Darkfield microscopy images of WT (M and N) and Y222F MTs (O and P). In N, the WT MTs in solution are shown, whereas in O and P, the Y222F MTs attached on the glass surface are shown. Because of the high MT densities, we could not record images of the Y222F MTs in solution using darkfield microscopy.

---

compared with the WT, suggesting that Y222F tubulin nucleates faster than WT tubulin (Oosawa and Asakura, 1975). The acceleration of nucleation was also confirmed by using darkfield microscopy. Immediately after the onset of the reaction (at 30 s), while the Y222F mutant produced many short MTs (initial tubulin concentration, 5 μM; Fig. 1 O), WT did not form any MTs (10 μM; Fig. 1 M). At 10 min, WT MTs were observed, but they were fewer in number and longer than the Y222F MTs (Fig. 1, N and P). Because of the very rapid nucleation of the Y222F mutant, 5 μM Y222F tubulin and 10 μM WT tubulin were the closest concentrations we could compare.

### Y222F mutation favors straight oligomers

In the described pair of experiments (5 μM Y222F and 10 μM WT), we analyzed the oligomers that coexisted with the MTs by using negative stain EM. While EM observation requires the sample to be at a low concentration, on the order of submicromolar or less (Mozziconacci et al., 2008), the use of the rapid flush method allowed us to quickly dilute the sample (<30 ms; Frado and Craig, 1992; Imai et al., 2015) and to capture images of the oligomers in the solution used for turbidimetry (Fig. 2, A–C; and Fig. S2). For both WT and Y222F tubulin, oligomers existed at the early stage of assembly when the turbidity was rising, but their numbers declined when the turbidity reached a plateau, suggesting the possibility that these oligomers might include on-pathway intermediates crucial for MT nucleation. The comparison between the GTP and GDP oligomers should provide information on the structural pathway of nucleation (Fig. 1 B).

As we measured the length and curvature of the WT and Y222F oligomers that appeared in the early stage of assembly (Fig. 2, D–G; sampling time, 5 min and 15 s, indicated by the white arrowheads in Fig. 1, I and J, respectively), both the WT and Y222F oligomers showed a broad range of curvatures, with the Y222F oligomers less curved than the WT oligomers (mean ± SD, 25.0 ± 12.4 and 16.9 ± 10.8 μm⁻¹ for WT and Y222F, respectively; Mann–Whitney $U$ test, P < 0.01). We also recorded the curvature distribution of oligomers assembled in the GDP condition (Melki et al., 1989; Valiron et al., 2010). The GDP oligomers were found to be more curved than the GTP oligomers for both WT and Y222F (mean ± SD, 34.4 ± 12.2 and 22.0 ± 11.8 μm⁻¹ for WT and Y222F, respectively; Mann–Whitney $U$ test, P < 0.01 in both WT and Y222F pairs). The distributions of the curvatures appeared similar to what was reported for the protofilaments at the growing ends of MTs (Guesdon et al., 2016; Orbach and Howard, 2019).

Most importantly, a subpopulation of nearly straight oligomers was observed in the presence of GTP (with a curvature

<10 μm⁻¹, highlighted in yellow in Fig. 2 G). The appearance of these near-straight oligomers correlates with the nucleation rate. With Y222F(GTP) tubulin showing rapid nucleation (Fig. 1 J), 31% of the oligomers were nearly straight, whereas with WT(GTP) tubulin showing moderate nucleation (Fig. 1 I), only 12% of the oligomers were nearly straight. For WT(GDP) tubulin incapable of nucleation, such oligomers composed <2% of the total.

Notably, 16% of the Y222F(GDP) oligomers were nearly straight (sampling time, 30 min), which is comparable to the probability observed for the WT(GTP) oligomers and well above that of straight WT(GDP) oligomers. Given that the T5 loop in Y222F(GDP) was not anchored in the "in" conformation (Fig. 1 G), this implies a possibility that Y222F(GDP) tubulin nucleates MTs. Indeed, incubating Y222F(GDP) tubulin in the condition of the turbidimetry assay led to a signal that slowly increased over time (Fig. 3, A–D). At 2 h after the onset of the reaction, using darkfield microscopy, we observed Y222F(GDP) MTs, which were substantially longer and fewer in number than the Y222F(GTP) MTs. Slow, steady elongation of Y222F(GDP) MTs was also confirmed in a MT dynamics assay (Fig. 3, E–K). Although GTP is required for canonical nucleation under natural conditions, the straight oligomers obtained with the mutated tubulin partially complemented the lack of γ-phosphate.

### The majority of single-stranded oligomers are below the critical size

The correlation between the proportion of straight oligomers and the nucleation rate indicates that straight oligomers are essential components of MT nucleation. How do these straight oligomers relate to the critical nucleus? Do they exceed the critical size to enter a stage where growth is thermodynamically favorable? To answer these questions, we calculated the size of the critical nucleus from the turbidity curves and found that the majority of the oligomers we analyzed in Fig. 2, E–G, were smaller than the critical size, as explained below.

To relate the tubulin concentration dependence of polymerization to the size of the critical nucleus, the turbidity (Fig. 1, I and J) was first converted to the amount of tubulin in the MTs (Fig. 4, A and B), taking the turbidity coefficient into account (see Materials and methods for details). The theory of nucleation predicts that this polymer mass should increase quadratically with time in the early stage of polymerization because both the number of MTs and the mass of individual MTs after nucleation increase linearly with time (Eq. 2 in Materials and methods). Our turbidity data fit well to the quadratic profile, supporting this view. The coefficient of the quadratic function should then be half of the nucleation rate, $I_0$, multiplied by the MT growth rate,

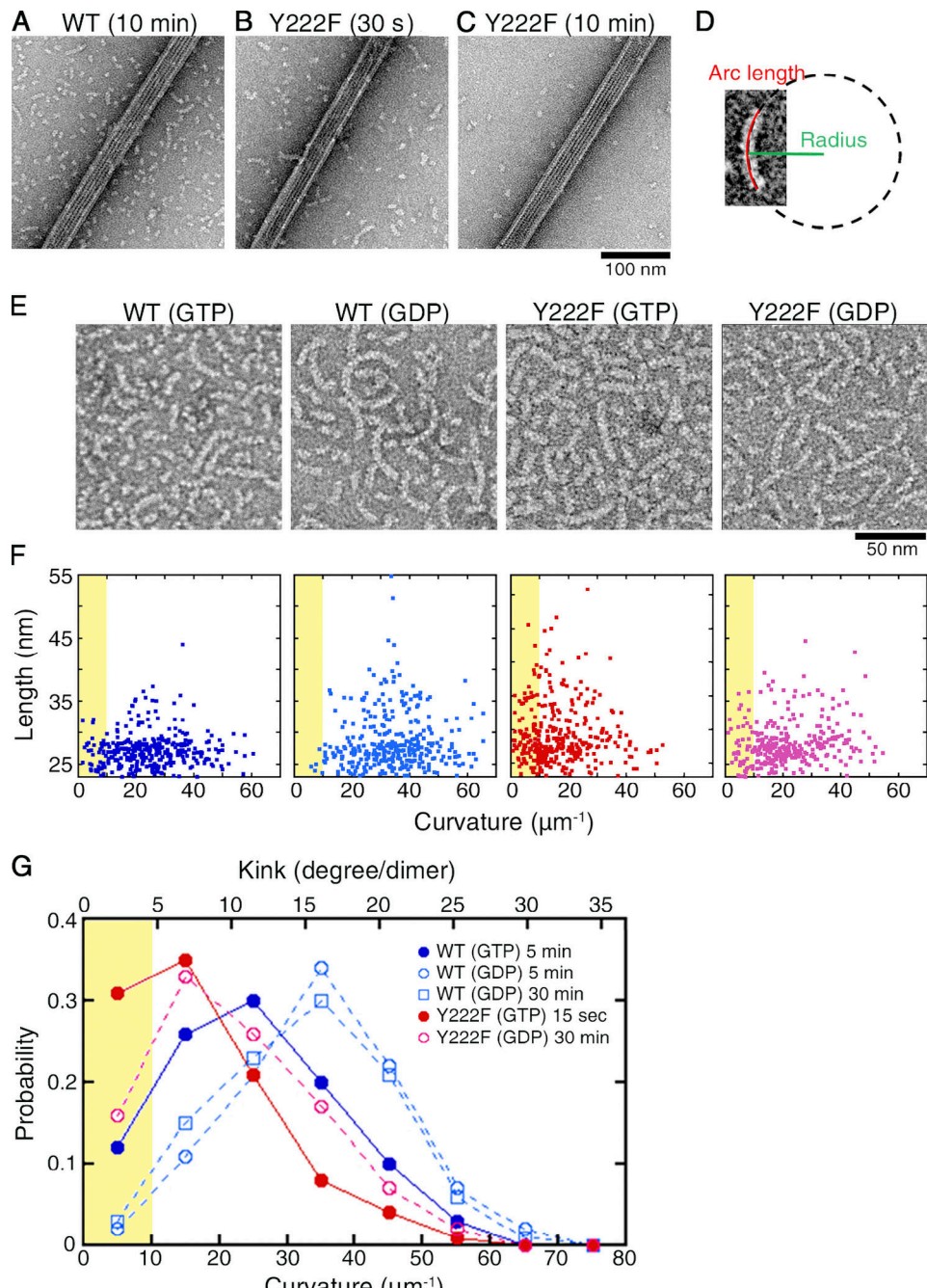

Figure 2. **Y222F mutation favors straight oligomers. (A–C)** Electron micrographs of oligomers coexisting with MTs. At 10 min after the onset of reaction, the EM image of WT tubulin showed dimers and oligomers coexisted with MTs (A). For the Y222F mutant, oligomers were observed at 30 s (B) but not at 10 min (C). For darkfield microscopy images of the same samples, see Fig. 1, M–P. **(D–G)** Statistical analyses of oligomers. Schematic representation of the measurement (D). Oligomer length and average curvature were determined by fitting a circle to the center line of the oligomer. To unambiguously determine the curvature, only those oligomers comprising at least three heterodimers were subjected to the analyses. **(E)** Representative images of oligomers and (F) distribution of the length and curvature for WT(GTP), WT(GDP), Y222F(GTP), and Y222F(GDP) oligomers ($n$ = 323, 321, 337, and 300, respectively). To show ensemble of oligomers, in the representative images (E), the oligomers were imaged at oligomer densities threefold to fivefold higher than the oligomer densities the curvature was measured. The exact condition of sample preparation is indicated in the graph legend (G). **(G)** The normalized distributions of the curvatures. For WT(GDP) oligomers, the grid was prepared at 5 and 30 min after the onset of reaction, but only the former is shown in E and F. The area corresponding to nearly straight oligomers with curvature <10 $\mu m^{-1}$ is highlighted in yellow in F and G.

$\nu_0$. Substituting $\nu_0$ with the actual growth rate (Fig. 3 K), we found that $I_0$ increases with the $\alpha$-th power of the initial tubulin concentration $C_0$, with $\alpha$ being 4.0 ± 0.2 and 5.9 ± 0.3 tubulin dimers (mean ± errors of fit) for WT and Y222F tubulin, respectively

(Fig. 4 C). $\alpha$ can be interpreted as the minimum size of the oligomers destined to grow, i.e., the size of the critical nucleus. Below this size, the oligomers are in quasi-equilibrium and are undergoing stochastic growth and dissociation. As we compared

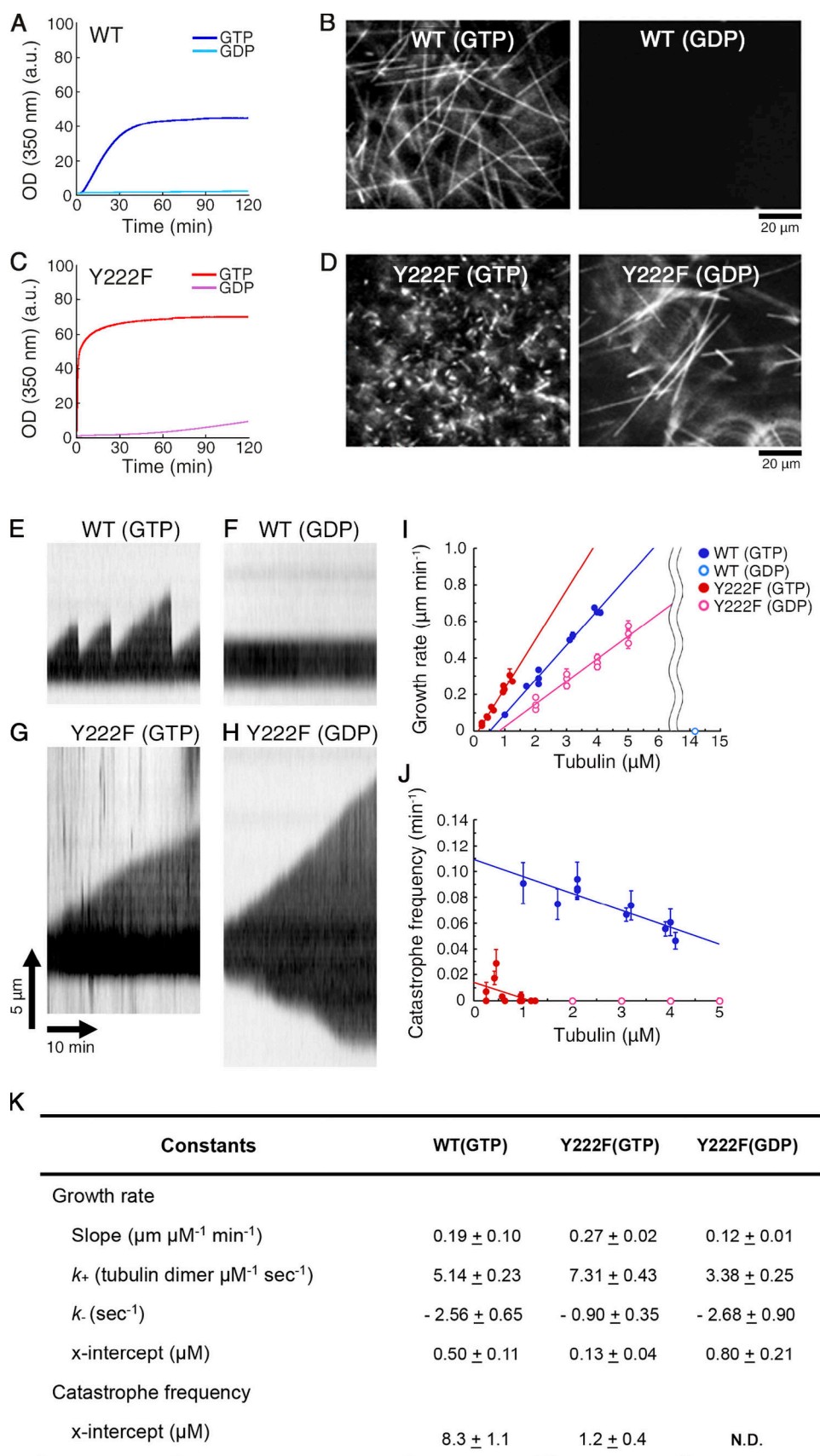

Figure 3. **Assembly of Y222F tubulin in the presence of GDP. (A and C)** Time course of polymerization of WT (10 µM; A) and Y222F tubulin (6 µM; C) monitored by OD$_{350}$. **(B and D)** Darkfield micrographs of WT MTs (B) and Y222F MTs (D), imaged at the end of reaction (120 min). While WT(GTP) and

Y222F(GTP) MTs were diluted for optimal observation of individual MT filaments, Y222F(GDP) MTs were observed without dilution. **(E–K)** Dynamic instability of individual WT and Y222F MTs. **(E and G)** Kymographs of WT (E) and Y222F (G) MTs polymerized in the GTP condition (tubulin concentrations of 2.1 and 1.3 µM, respectively). **(F and H)** Kymographs of WT (F) and Y222F (H) MTs polymerized in the GDP condition (8.4 and 3.0 µM, respectively). **(I and J)** Concentration dependence of the growth rate (I) and catastrophe frequency (J). In the case of Y222F(GDP), the growth rate and the catastrophe frequency at the plus end are reported. In I, the regression line for WT(GTP), Y222F(GTP), and Y222F(GDP) can be represented by the equations $y = 0.19 x – 0.09$ (R = 0.99), $y = 0.27 x – 0.03$ (R = 0.98), and $y = 0.12 x – 0.10$ (R = 0.97), respectively. Error bars represent SEM. Total number of datasets was 966, 137, and 352 for WT(GTP), Y222F(GTP), and Y222F(GDP), respectively. In J, total time of elongation per data point was 112–2,491 min, with error bars showing SD. The regression line for WT(GTP) and Y222F(GTP) can be represented by the equations $y = –0.013 x + 0.109$ (R = 0.88) and $y = –0.012 x + 0.014$ (R = 0.46), respectively. **(K)** Kinetic and thermodynamic parameters calculated from the data shown in I and J. Values represent the mean ± SD. For details of statistical analysis, see Materials and methods.

these sizes with the actual length of the oligomers (Fig. 4 D and Fig. 5, A–F), for both WT and Y222F tubulins, a very large majority of the oligomers (>99%) was found to be below the critical size. Our results collectively imply that it is the straight oligomers of the size α that are critical nuclei for MT assembly.

From the length distribution of the subcritical oligomers, we also calculated the free energy change associated with tubulin binding to an oligomer, $\Delta G_{olig}$ (Fig. 5, D–G; see Materials and methods for the calculation). The results showed that the interdimer interaction is stronger in the Y222F oligomer than that in the WT oligomer by

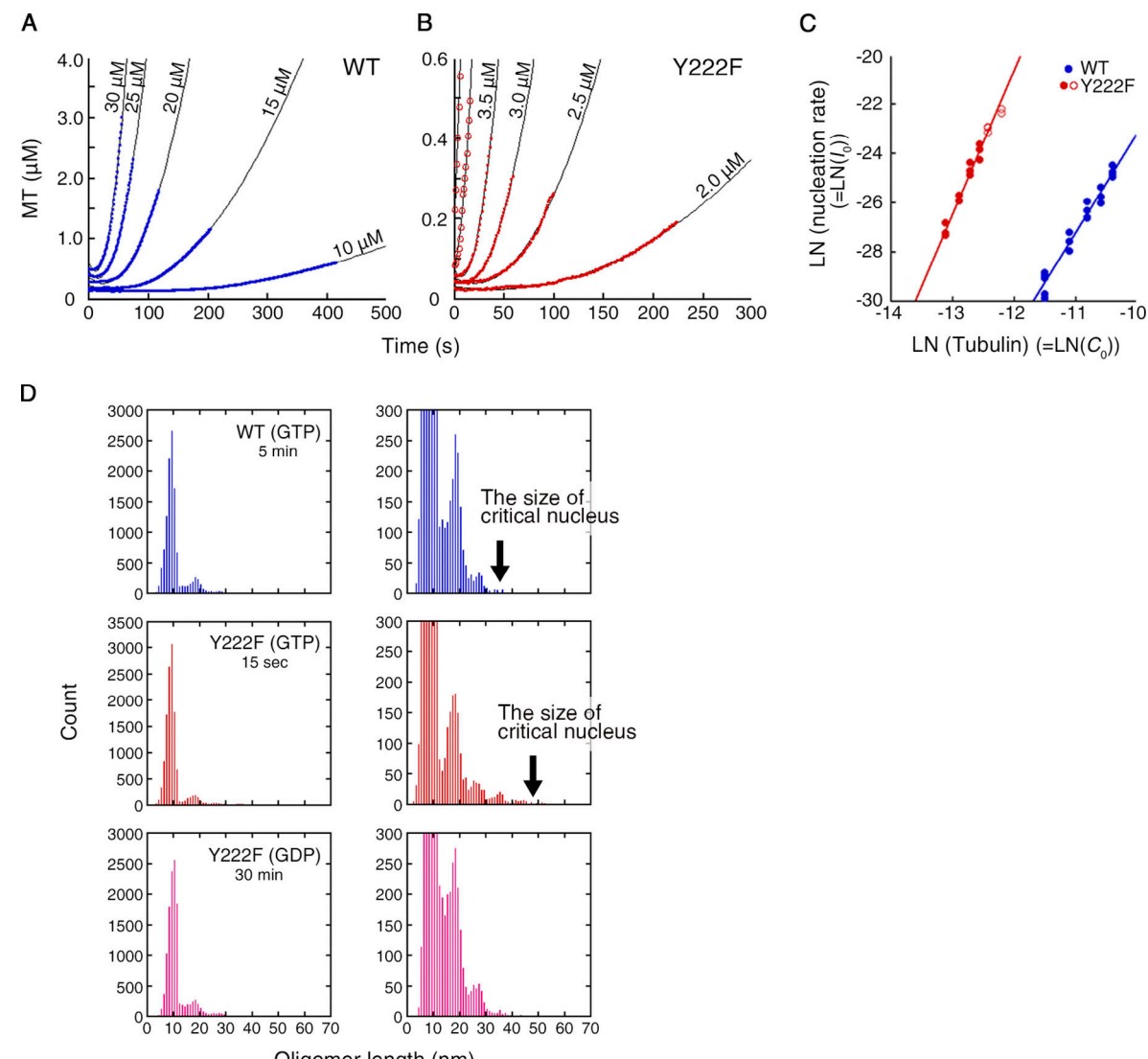

Figure 4. **The majority of single-stranded oligomers are below the critical size. (A and B)** Time course of the polymerization of WT (A) and Y222F tubulin (B) from time zero to $T_{10\%}$, which was fit by a quadratic function of time (R > 0.99 for all curves). The original data are shown in Fig. 1, I and J. **(C)** Log–log plots of the rate of nucleation, $I_0$, vs. the initial tubulin concentration, $C_0$, resulted in a linear line with the slope (size of critical nucleus) of 4.0 ± 0.2 and 5.9 ± 0.3 for WT and Y222F, respectively. For the Y222F mutant, the data measured at 4 and 5 µM tubulin (red open circles) were excluded from the calculation (see Materials and methods for details). **(D)** The distributions of the oligomer length in the experiment shown in Fig. 2, E–G. n = 11,527, 12,606 and 12,430 for WT(GTP), Y222F(GTP), and Y222F(GDP), respectively. In the right panel of each pair, the scale of the vertical axis is enlarged.

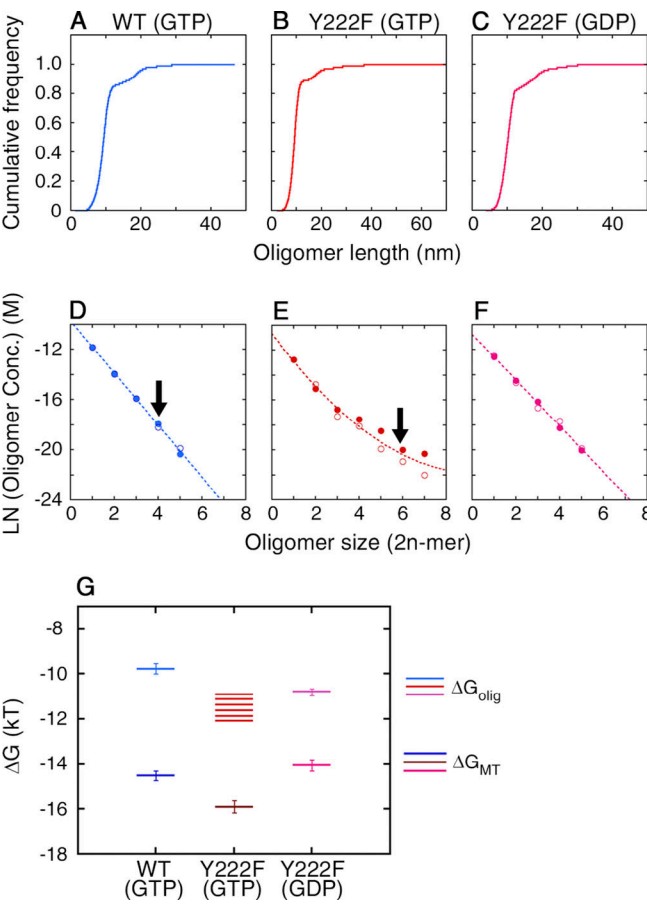

Figure 5. **Size distribution of oligomers. (A–C)** Cumulative frequency of the length, calculated from the length distribution presented in Fig. 4 D. **(D–F)** Logarithm of the concentration of the oligomers, $x_{2n}$, calculated from the probability of oligomers with different sizes (see Materials and methods for the details of the calculation). The measurement was made twice for each type of tubulin (represented by filled and open circles). The raw data for the first round of measurement are shown in Fig. 4 D. For the second round of measurement, $n$ = 10,945, 11,803, and 11,092 for WT(GTP), Y222F(GTP), and Y222F(GDP), respectively. For both WT(GTP) and Y222F(GTP) tubulins, the largest oligomer was only one unit larger than the size of the critical nucleus, indicated by arrows. The data for the WT(GTP; D) and Y222F(GDP) oligomers (F) showed the exponential decay of the oligomer concentration with size, which can be fit by the equations y = −9.75 − 2.07 x (R = 0.99) and y = −10.79 − 1.83 x (R = 0.99), respectively. In the case of the Y222F(GTP) oligomers (E), the concentration of oligomers with $n > 3$ was significantly higher than the concentration expected from simple exponential decay and was best fit by the equation y = −10.65 − 2.36 x + 0.12 $x^2$ (R = 0.98). **(G)** The free energy change associated with the binding of tubulin dimer to oligomer ($\Delta G_{olig}$) and MT ($\Delta G_{MT}$). The error bars for each $\Delta G_{MT}$ represent the errors estimated from the SD of $k_+ / k_-$ (see Materials and methods for details of calculation). Conc., concentration.

1–2 $k_BT$ ($\Delta G_{olig}$ = −9.8 ± 0.2 $k_BT$ and in the range of −10.7 to −12.1 $k_BT$ for WT(GTP) and Y222F(GTP) oligomers, respectively). The value of $\Delta G_{olig}$ for WT is similar to the values reported for brain tubulin (Erickson and Pantaloni, 1981; Gardner et al., 2011; VanBuren et al., 2002). In the Y222F mutant, the interdimer interaction might have been strengthened by the modulation of the T5 loop.

### Oligomers above the critical size form multi-stranded complexes

In the above experiment with Y222F(GTP) tubulin, we rarely observed double- or triple-stranded oligomers among the vast majority of single-stranded oligomers (Fig. 6, A–J). In the double-stranded oligomers, the length of the longer strand was >50 nm (Fig. 6 L), clearly exceeding the size of the critical nucleus, and also the size of the majority of single-stranded oligomers (Fig. 6 K). The shorter strands had lengths both above and below 50 nm (Fig. 6 L). These results indicate that a single-stranded oligomer that reaches the size of the critical nucleus may laterally associate with a dimer or another oligomer, and continue to grow as a multi-stranded oligomer. In a later stage (30 s or longer), we observed sheets composed of a few protofilaments that were several hundred nanometers in length, likely on their way to becoming MTs (Fig. 7; Roostalu and Surrey, 2017; Voter and Erickson, 1984).

We could not find multi-stranded oligomers of WT tubulin (10 µM at 5 min). It is not surprising that we could not find such WT oligomers, considering that the nucleation rate of WT tubulin was approximately three orders of magnitude smaller than that of Y222F tubulin (the nucleation rates in Fig. 4 C were 2.1 × $10^{-10}$ and 2.1 × $10^{-13}$ M/s for 5 µM Y222F and 10 µM WT tubulin, respectively). Theoretically, at higher tubulin concentrations, we may be able to see multi-stranded WT oligomers, but the experiment in such conditions was difficult because of the aggregates. Under conditions where nucleation does not occur, for example, in tubulin solution incubated at 4°C (Fig. S3), we never observed double- or triple-stranded oligomers.

### The Y222F mutation suppresses catastrophes

Y222F tubulin favoring a straight conformation is more efficient than WT tubulin not only in nucleation but also in elongation of MTs. As we compared the dynamic instability of WT and Y222F MTs in the GTP condition, the Y222F mutant showed a reduced catastrophe frequency and higher growth rate (Fig. 3, E–K). Y222F(GDP) tubulin capable of nucleation was also capable of elongation, although at a rate lower than that of Y222F(GTP) or WT(GTP) tubulin.

From the concentration dependence of the growth rate, we calculated the free energy change associated with the binding of tubulin to the end of an MT, $\Delta G_{MT}$ (Fig. 5 G; see Materials and methods for calculation). In both WT(GTP) and Y222F(GTP), tubulin was ~4 $k_BT$ more stabilized upon integration into the MT lattice than upon binding to the end of a single-stranded oligomer ($\Delta G_{olig}$). This difference is likely caused by the energy gain due to the lateral interaction, as was reported for brain tubulin (Gardner et al., 2011; VanBuren et al., 2002).

## Discussion

We demonstrate here the linkage between GTP-dependent movement of the T5 loop, formation of straight oligomers of critical size, and nucleation of MTs. Our novel approach using recombinant tubulin and rapid flush negative stain EM allowed us to identify the structural pathway of MT nucleation in vitro.

### Structural pathway of GTP-dependent nucleation

By linking the statistical analysis of pre- and post-critical nuclei with the kinetic analysis of MT assembly, we propose the following scheme for GTP-dependent nucleation of MTs. The

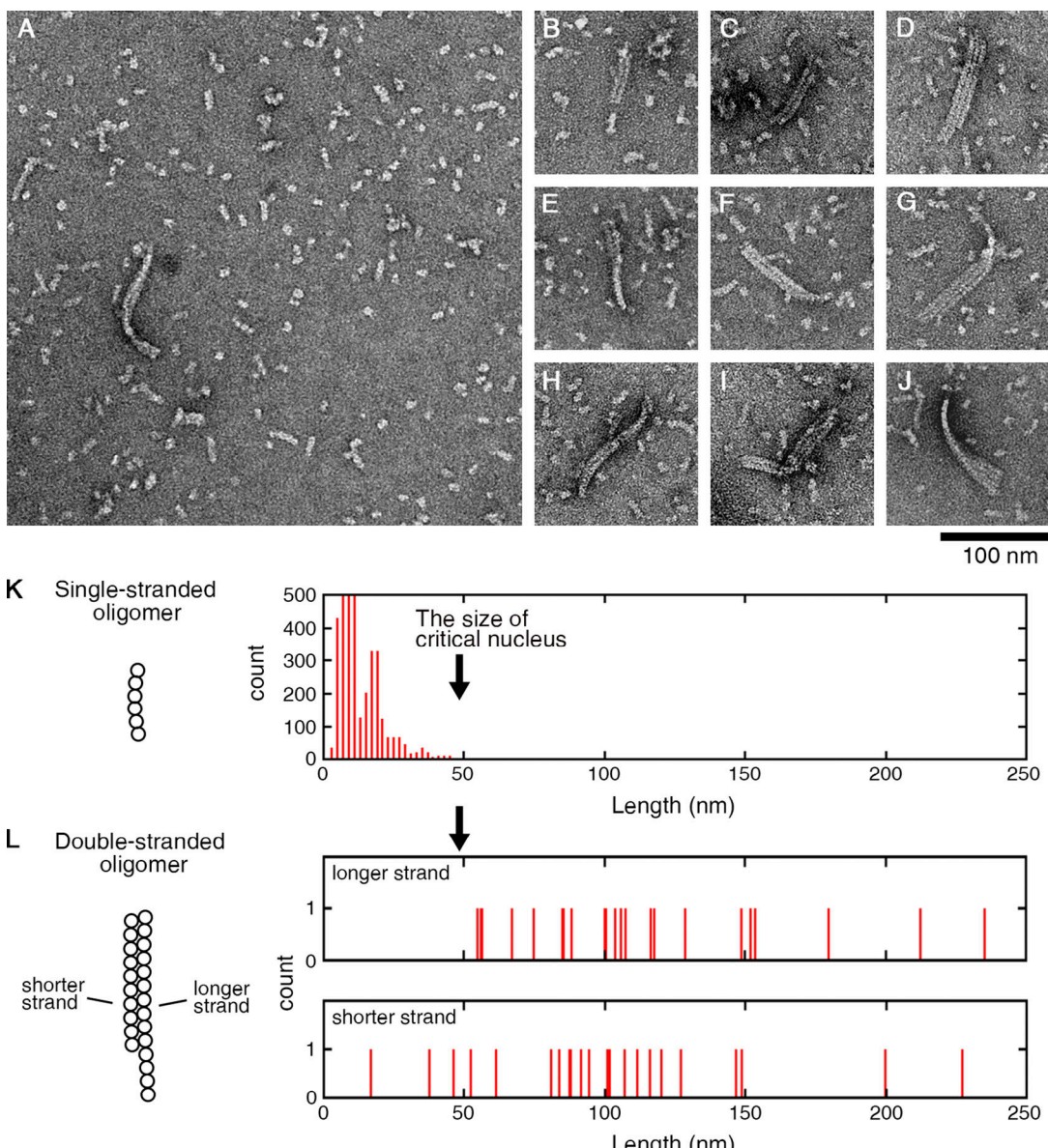

Figure 6. **Y222F oligomers above the critical size form multi-stranded complexes. (A)** Among the majority of dimers and single-stranded oligomers, multi-stranded oligomers were rarely found. **(B–J)** A catalog of two- or three-stranded oligomers observed at 15 s after the onset of reaction. The multi-stranded oligomers with larger size were found in later stage (see Fig. 7). **(K)** Length of single-stranded oligomers (identical to Fig. 4 D). **(L)** Length of the longer and shorter strands in each of the two-stranded oligomer complex (see Materials and methods for details of measurement).

dimers and sub-critical oligomers of variable sizes and curvatures are in quasi-equilibrium in tubulin solution (Fig. 8 A). The GTP-dependent extension of the T5 loop mediates the establishment of longitudinal interdimer contacts and stabilizes them (Natarajan et al., 2013; Nawrotek et al., 2011), giving rise to a subpopulation of nearly straight oligomers (highlighted in yellow in Fig. 2 G). Once these contacts are established, the T5 loop may switch back to an "in" conformation, as seen in the MT core (Zhang et al., 2018). The oligomers also interact laterally and transiently to form double-stranded oligomers, but most of them dissociate almost immediately because of the weakness of the lateral interactions compared with the longitudinal ones (Manka and Moores, 2018; Mickolajczyk et al., 2019; Nogales et al., 1999). With increasing oligomer size, the

population decays exponentially (Fig. 4 D and Fig. 5 D), whereas the potential to form double-stranded oligomers increases owing to the increased lateral interface. At the size of critical nucleus, the differential entropic cost for gathering tubulin dimers into a straight oligomer should balance with the differential energy gain due to the lateral association, allowing the formation of two-stranded oligomers that serve as a platform for the growth of protofilaments (seed; Fig. 8 B, lower panel).

The binding of a tubulin molecule to such a multi-stranded straight oligomer might be thermodynamically more favorable than binding to a single-stranded oligomer, because in the former case, simultaneous longitudinal and lateral interactions are possible. Our analyses showed that the free energy gain associated with the

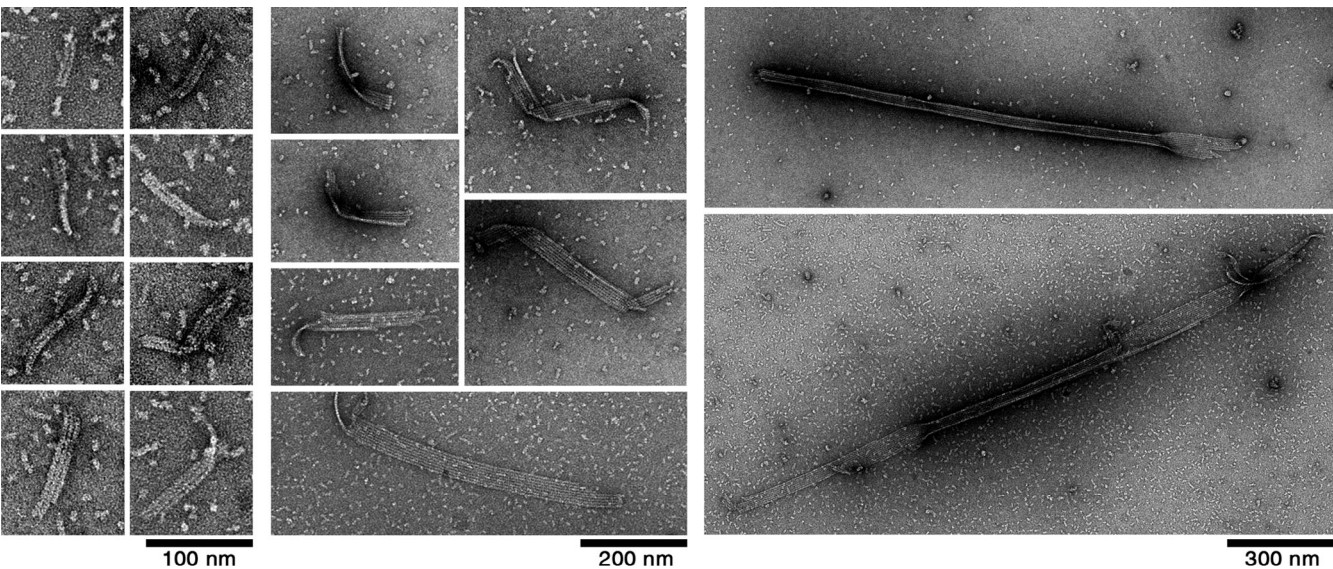

Figure 7. **EM images of the intermediate structures in the Y222F mutant.** Multi-stranded oligomers and sheets observed at 30–120 s after the onset of the reaction.

tubulin binding to a MT is 4 $k_B T$ larger than the free energy gain associated with the binding to a single-stranded oligomer (Fig. 5 G). Our values, $\Delta G_{olig}$ and $\Delta G_{MT}$, are consistent with the values reported by other groups (Erickson and Pantaloni, 1981; Gardner et al., 2011; VanBuren et al., 2002). Similarly, tubulin that binds to a multi-stranded oligomer would gain an energy equivalent to this 4 $k_B T$

(Fig. 8 B, lower panel). For a multi-stranded oligomer composed of curved oligomers, if any, such a thermodynamic stabilization is difficult because an incoming tubulin would participate only in the longitudinal interaction (Fig. 8 B, upper panel). According to this scheme, nucleation depends on a very rare event, i.e., formation of a straight oligomer with critical size.

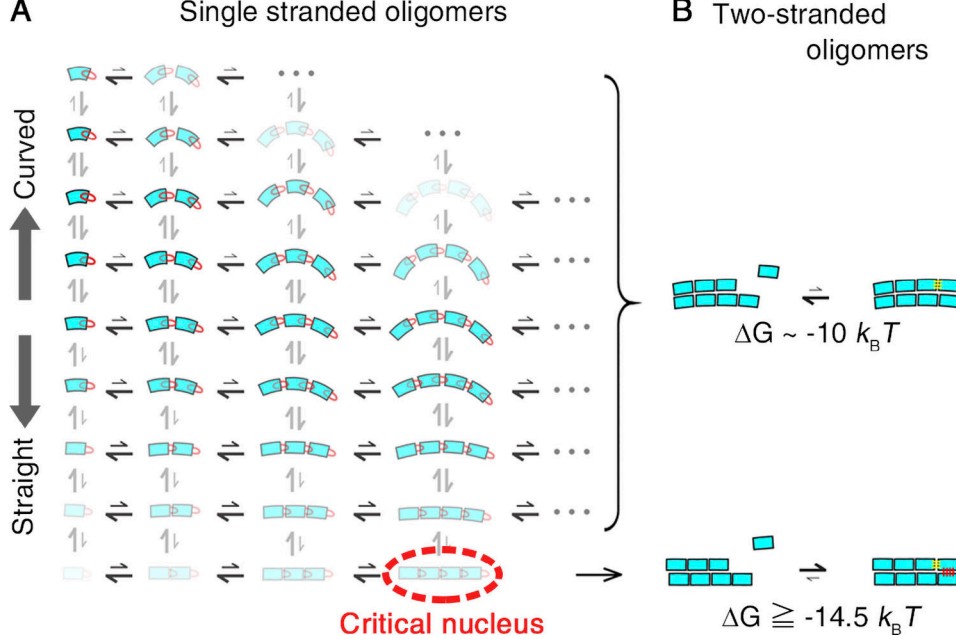

Figure 8. **A model for the structural pathway of spontaneous nucleation. (A)** In a solution of GTP tubulin, tubulin dimers and oligomers of variable sizes and curvatures are in rapid equilibrium. The color intensity represents the probability for each dimer and oligomer. The distance between tubulin dimers in an oligomer symbolically represents the strength of the interdimer bonds. Straight oligomers crucial for nucleation can be formed only from GTP tubulin, the T5 loop "out" conformation being favored. **(B)** While the reaction of tubulin binding to a two-stranded oligomer composed of straight protofilaments can be thermodynamically favorable (bottom), the reaction of tubulin binding to a two-stranded oligomer formed from curved protofilaments is not (top). Here, we assume that a tubulin dimer that binds to a multi-stranded oligomer gains energy equivalent to the energy gained by a tubulin dimer that binds to a MT protofilament ($\Delta G_{MT}$).

Here we assume the WT critical nucleus to enter a phase of lateral association as the Y222F critical nucleus does, although we observed the multi-stranded oligomers only with the Y222F mutant (Fig. 6). It is reasonable to expect the WT critical nuclei to form multi-stranded complexes because for the structural intermediates to overcome the energy barrier for nucleation, the dimension of the growth has to be shifted from one to two dimensions, allowing the coming subunit to make a higher number of bonds with the nucleus. In the case of actin nucleation, thermodynamically unfavorable assembly of filament turns thermodynamically favorable when the filament reaches a critical size (tetramer), where the growth is shifted from one to two dimensions (Oda et al., 2016; Oosawa and Asakura, 1975). The validity of our model needs to be confirmed in future experiments verifying the correlation between the nucleation rate and oligomer curvature/length in different experimental conditions.

**Earlier approaches to MT nucleation and our innovations**

Voter and Erickson (1984) estimated the size of the critical nucleus in spontaneous nucleation to be 6–7, based on the tubulin concentration dependence of the nucleation rate. Others reported a similar (Caudron et al., 2000) or larger value (Kuchnir Fygenson et al., 1995) as a size of the critical nucleus, with the origins of differences attributable to the experimental conditions and to the method of analysis.

For a pathway of MT nucleation, Voter and Erickson (1984) assumed that the critical nucleus is a two-stranded oligomer and that, for complete nucleation, lateral association of tubulin molecules to form a third strand is required. They fit the entire course of the turbidity curve using this two-step model, but their theoretical curves do not seem to match the turbidity data, especially in the early phase where nucleation happens (for example, see Fig. 6 in Voter and Erickson, 1984). In contrast, in our kinetic analyses, we used the turbidity data only in the early phase ($t < T_{10\%}$) and fit them by a simple model assuming one rate-limiting step, which yielded a very good correlation between theoretical curves and the raw data (Fig. 4, A and B). Our analyses also clarified that the initial delay in the increase of turbidity after the onset of reaction, formerly regarded as lag (Johnson and Borisy, 1977), is not a lag but a part of a parabolic curve.

The involvement of oligomers or ribbon structure in nucleation has been implicated in a time-resolved x-ray scattering study (Spann et al., 1987) and in negative stain EM studies (Mozziconacci et al., 2008; Portran et al., 2017; Wang et al., 2005). In these earlier works, to image individual oligomers/ribbons, the experiment was conducted either at a low tubulin concentration or at a low temperature where MT nucleation did not happen. The use of rapid flush negative stain EM allowed us to visualize the nucleation intermediates on the canonical pathway of nucleation. The correlation between the oligomer curvature/length and the nucleation rate was systematically analyzed for the first time in this study.

**The mechanism of accelerated nucleation by Y222F mutation**

The nucleation rate of Y222F tubulin was approximately three orders of magnitude higher than that of WT tubulin ($2.1 \times 10^{-10}$ and $2.1 \times 10^{-13}$ M/s for 5 µM Y222F and 10 µM WT tubulin, respectively;

Fig. 4 C), despite the lower concentration of Y222F oligomers compared with the WT counterparts (0.4 and 1.0 µM for Y222F and WT, respectively; the sum of the concentration of oligomers, except for dimers, is shown in Fig. 5, D and E). Therefore, individual Y222F oligomers should have higher ability to become an MT compared with the WT counterpart. We cannot know the exact curvature of the critical nuclei, but if we assume the oligomer curvature to be independent of the size, Y222F critical nuclei should be straighter than the WT critical nuclei (Fig. 2, F and G). In short, the Y222F mutation caused a stronger inter-dimer bond (Fig. 5 G), leading to a longer, straighter critical nucleus. Because lateral interaction between oligomers is reliant on the electrostatic attractive force between oligomer interfaces (Manka and Moores, 2018; Nogales et al., 1999), longer Y222F oligomers may be more advantageous for lateral interaction, leading to higher nucleation rate compared with WT oligomers. In other words, the low energetic cost of gathering Y222F tubulin allowed this mutant to find a balance between the entropic cost and energy gain at a larger size than the WT.

We do not know the molecular mechanism by which the mutation causes longer straighter oligomers. Notably, the inter-dimer interaction appears to be strengthened in a cooperative manner in this mutant. While in the WT, the concentration of WT oligomer decays exponentially with size, the concentration of Y222F oligomer deviates upward from the exponential distribution, indicating a progressive stabilization with size (Fig. 5, D, E, and G). One possibility is that in the mutant, the local structure around the T5 loop could change with oligomer size, allowing higher stability for larger oligomers. Although tubulin structures within the $T_2R$ complex showed the T5 loop in the same "out" conformation with GTP in both WT and Y222F mutants (Fig. 1, F and H), the situation could be different in oligomers. Alternatively or additionally, one can hypothesize that during the assembly of oligomers, WT oligomers gradually increase their curvature because of GTP hydrolysis (Carlier et al., 1997). The effect of GTP hydrolysis could be less pronounced in the Y222F mutant, as the T5 loop does not switch back to the "in" conformation. This hypothesis may also explain the high nucleation efficiency reported for a tubulin mutant unable to hydrolyze GTP (Roostalu et al., 2020) or for GMPCPP tubulin (Hyman et al., 1992), but the experimental proof for the hypothesis is yet to come.

The residue β-Y222 is highly conserved in animals and fungi, probably because the slightest modification of this residue, such as tyrosine to phenylalanine, has a significant effect on nucleation (Table S2; Orbach and Howard, 2019). In *Homo sapiens*, tyrosine is conserved among all eight β-tubulin genes. When the tyrosine is replaced with phenylalanine in the *TUBB* gene, it leads to the congenital developmental disorder termed circumferential skin creases Kunze type (Isrie et al., 2015). Children with this disease have ring-like symmetrical folds of skin on the limbs with variable additional features such as intellectual disability and facial dysmorphism.

**Conformational diversity and selection**

There have been debates over the possible role of GTP/GDP in the induction of conformational change in tubulin. A simple allosteric model postulates that GTP binding causes straightening of tubulin required for the assembly of an MT. Contrary to this assumption, the small-angle x-ray scattering profiles

showed that unassembled tubulin adopts a curved conformation in solution regardless of the species of bound nucleotide (Manuel Andreu et al., 1989; Rice et al., 2008). In addition, small molecules that only bind to curved tubulin have similar affinities for GDP and GTP tubulin (Barbier et al., 2010). The lack of evidence for a nucleotide-dependent structural difference in tubulin has led some researchers to propose a "lattice model," in which the mechanical constraints during lateral assembly induce a structural change in tubulin (Rice et al., 2008). According to the latter model, GTP plays only a secondary role to tune the strength of longitudinal contacts. It should be noted that, in earlier studies, the structures of GDP and GTP tubulin were compared under conditions where polymerization does not occur (Manuel Andreu et al., 1989; Rice et al., 2008; Nawrotek et al., 2011; Pecqueur et al., 2012). By contrast, we compared the GDP vs. GTP oligomers in conditions where the MT nucleation actually occurs for the latter. This choice of experimental condition and the use of rapid flush negative stain EM allowed us to detect the GTP-dependent straightening in oligomers. Our results demonstrate that GTP is indeed an allosteric effector responsible for tubulin (oligomer) straightening.

GTP binding causes just a small shift in the broad spectrum of tubulin oligomer conformations (Fig. 2 G), not a transition between discrete "curved" and "straight" conformations, as anticipated in earlier studies. Diversity and selection are key factors controlling nucleation. The suppression of catastrophe in polymerization by the Y222F mutation (Fig. 3, J and K) indicates that diversity and selection might also be the keys to control polymerization/depolymerization. In the case of native brain tubulin, the oligomers/protofilaments show a broad spectrum of curvature, and the binding of GMPCPP and taxol increased the population of straight oligomers/protofilaments (Elie-Caille et al., 2007; Müller-Reichert et al., 1998). This increase in the population of straight oligomers/protofilaments parallels the acceleration of MT nucleation and the inhibition of catastrophe (Díaz and Andreu, 1993; Müller-Reichert et al., 1998; Sandoval and Weber, 1980; Schiff et al., 1979; Schiff and Horwitz, 1980). It is possible that any factor that stabilizes straight oligomers/protofilament accelerates nucleation, and later in the growing phase facilitates polymerization and prevents catastrophe at the growing end of MTs. In both cases, stable growth of oligomer/protofilament may require the coming tubulin to simultaneously make longitudinal and lateral bonds with an array of straight oligomers/protofilaments (Fig. 8, lower panel; Erickson and Pantaloni, 1981; Mickolajczyk et al., 2019).

In cells, MTs are nucleated at a tubulin concentration significantly lower than those required for nucleation in vitro. High efficiency of nucleation owes to the template of γTuRC and to MT-associated proteins (MAPs) that promote MT growth (e.g., XMAP215; Flor-Parra et al., 2018; Thawani et al., 2018; Wieczorek et al., 2015; Woodruff et al., 2017) or suppress catastrophes (e.g., TPX2; Roostalu et al., 2015; Wieczorek et al., 2015; Woodruff et al., 2017). The γTuRC-mediated nucleation shows a power law dependence on tubulin concentration (Zheng et al., 1995; Fig. S4), suggesting that templated nucleation also requires tubulin to assemble to a critical size. Whether these activities of MAPs are attributable to the stabilization of straight oligomers/

protofilaments remains to be determined. The methods of analysis developed in this study will be useful to test such hypotheses.

## Materials and methods

### Construction of baculovirus transfer plasmids

*Drosophila* α1-tubulin (*αTub84B*; NT_033777) and β1-tubulin (*βTub56D*; NT_033778) genes were obtained from the Drosophila Genomics Resource Centre. For affinity purification, tags were fused to the 3' end of each clone (for α1-tubulin, a His$_8$ tag and a sequence encoding FactorXa cleavage site [IEGR] were linked by a glycine-based linker sequence [GGSGG], and for β1-tubulin, a FLAG tag and an IEGR were linked by a linker sequence [GGG]). To increase the expression level, an L21 leader sequence was also added just before the start codon (Sano et al., 2002). To exclude variability due to acetylation, the α1 was made unacetylatable by residue substitution K40R, and it was treated as WT. The inserts were cloned into the pFastBac Dual vector (Life Technologies; Minoura et al., 2013).

### Purification of recombinant tubulin

The recombinant tubulin was expressed in HighFive cells (Life Technologies) and purified by three steps of column chromatography (diethylaminoethyl sepharose ion exchange chromatography, His-affinity column, FLAG-affinity column), as previously described (Minoura et al., 2013), with the exception of the following. While 1 mM GTP was included in solutions used in all three steps of the purification of WT tubulin, GTP was replaced by 1 mM GDP for purification of Y222F tubulin (Fig. S1). The tubulin eluted from the FLAG-affinity column was concentrated to >5 mg/ml with an Amicon Ultracel-30K filter (EMD Millipore), centrifuged at 100,000 rpm (∼540,000 *g*; Beckman TLA-100.3 rotor) for 10 min at 4°C, and the supernatant was incubated at 30°C after adding glycerol (final 33% vol/vol) and GTP (2 mM). After 2 h of incubation, Factor Xa protease (New England Biolabs) was added to the MT solution at a molar ratio of 1:22 (Factor Xa:tubulin) and reacted overnight at 25°C. The next morning, the sample solution was centrifuged at 100,000 rpm (Beckman TLA-100.3 rotor) for 15 min at 30°C, and the precipitated MTs were suspended in BRB80 buffer solution (80 mM Pipes, 2 mM MgCl$_2$, and 1 mM EGTA, pH 6.8) containing 1 mM GTP (for WT tubulin) or 1 mM GDP (for Y222F tubulin) at 4°C, and left for 30 min on ice. To remove aggregates, the solution was centrifuged again at 100,000 rpm (∼430,000 *g*; Beckman TLA-100 rotor) for 10 min at 4°C. The supernatant was frozen in aliquots in liquid nitrogen, and they were stored at −80°C until use. The yield of tag-free tubulin was ∼3 mg from a 1-liter culture of HighFive cells (∼20 g of cells). Protein concentration was determined using a Pierce 660-nm Protein Assay Kit (Thermo Fisher Scientific).

### Preparation of GTP tubulin for turbidimetry and negative stain EM

To follow the preequilibrium dynamics upon nucleation of MTs (Figs. 1, 2, 3, 4, 5, 6, and 7), rigorous control of the starting material was important. To make sure that at time zero the

oligomers were in quasi-equilibrium with the dimers at 4°C, the tubulin sample was prepared fresh for each run of the experiment by the following protocol. Soon after the frozen tubulin sample was defrosted, it was filtered by an Ultrafree-MC VV Centrifugal filter (0.1-µm pore size; EMD Millipore) to remove aggregates, and then spun through a Micro Bio-Spin P30 Column (Bio-Rad Laboratories) to replace the solution by BRB80 buffer containing 1 mM GTP. A 70-µl tubulin fraction eluted from the column was centrifuged at 100,000 rpm (Beckman TLA-100 rotor) for 15 min at 4°C to remove aggregates and diluted at the appropriate concentration. The tubulin sample was used for turbidimetry and negative stain EM within 30 min after the ultracentrifugation (Fig. S3). The nucleotide content in the exchangeable site (E-site) was >95% GTP (Fig. S1 B).

### Preparation of GDP tubulin for turbidimetry and negative stain EM

Soon after being defrosted, tubulin was first filtered by an Ultrafree-MC VV Centrifugal filter (0.1-µm pore size; EMD Millipore) to remove aggregates. For Y222F tubulin, the sample was further spun through a Micro Bio-Spin P30 Column (Bio-Rad Laboratories) to replace GDP with GTP. WT and Y222F tubulins, thus conditioned, were incubated at 30°C for 2 h for polymerization. The sample solution was centrifuged on a 60% glycerol cushion at 100,000 rpm (Beckman TLA-100 rotor) for 15 min at 30°C to precipitate the MTs. After careful washing of the wall of the centrifuge tube and the pellet surface by nucleotide-free BRB80 buffer, the pellet was suspended in nucleotide-free BRB80 buffer at 4°C. The sample was left on ice for 30 min, and a 70-µl sample solution was centrifuged at 100,000 rpm (Beckman TLA-100 rotor) for 15 min at 4°C. The supernatant diluted at the appropriate concentration was used for turbidimetry and for preparation of the grid for negative stain EM. The nucleotide content in the E-site is ~100% GDP for both WT and Y222F tubulin (Fig. S1 B). Similar to the preparation of GTP tubulin, the sample was prepared fresh for each run of the experiment.

### Nucleotide content analysis

The tubulin sample was spun twice through a Micro Bio-Spin P30 Column, equilibrated with nucleotide-free BRB80, and denatured with trifluoroacetic acid (final 2%). After removal of the denatured protein by centrifugation, the supernatant containing the released nucleotide was filtered through an Ultrafree-MC GV Centrifugal Filter (0.22-µm pore size; EMD Millipore) to remove small debris. The filtrate was analyzed by fast protein liquid chromatography (AKTApurifier; GE Healthcare Life Sciences) using a MonoQ column (5/50 GL; GE Healthcare Life Sciences) equilibrated with 20 mM Tris-HCl, pH 8.0. The nucleotide was eluted using a 0–300 mM NaCl gradient. GDP, GTP, and GMPCPP solutions were injected as references to calibrate the column.

### CryoEM sample preparation

The WT and Y222F MTs were assembled at 15 µM and 4 µM of the WT and Y222F tubulin dimers in BRB80 buffer containing 1 mM GTP at room temperature (25–30°C), respectively. The WT

and Y222F MTs were applied to lacey carbon grids (Agar Scientific AGS166-4 400 mesh copper grids) and holey carbon grids (Quantifoil R1.2/1.3 300 mesh copper grids) immediately after application of glow discharge to the grids, and were vitrified by using EM GP (Leica) and Vitrobot (Thermo Fisher Scientific), respectively. The grids of the WT and Y222F MTs were observed at liquid nitrogen temperature by using a JEM 2100F electron microscope (JEOL) operated at 200 kV and a Tecnai Arctica electron microscope (Thermo Fisher Scientific) operated at 200 kV, respectively. The EM images of the WT and Y222F MTs were captured by an UltraScan 4000 CCD camera (GATAN) at nominal magnification of 50,000 with 0.213 nm/pixel and by a Falcon II direct electron detector (Thermo Fisher Scientific) at nominal magnification of 39,000 with 0.285 nm/pixel, respectively. The defocus range was between –2.6 µm and –3.5 µm. The total dose was 1,000–4,000 $e^-$/nm$^2$.

### Crystallization and structure determination

Crystals of *Drosophila* tubulin were obtained as $T_2R$ complexes and after seeding (Nawrotek et al., 2011). Such complexes are composed of two tubulin heterodimers and one stathmin-like domain of the RB3 protein (RB3$_{SLD}$). To avoid complications resulting from GTP hydrolysis during crystallization experiments, in the case of WT tubulin, GMPCPP was used as a surrogate of GTP. It has been shown with mammalian brain tubulin that GMPCPP and GTP tubulin share the same conformation (Nawrotek et al., 2011). However, in the case of the Y222F mutant, we did not succeed in obtaining crystals from GMPCPP-bound tubulin. Therefore, we determined instead the structure of Y222F(GTP) tubulin, from crystals harvested ~15 h after setting up the crystallization drops, a compromise between crystal size and GTP hydrolysis. The electron density maps suggested a full GTP occupancy at the β2 nucleotide-binding site and about half of GTP that has been hydrolyzed at the β1 nucleotide-binding site (see Fig. 1 C for subunit nomenclature). Data for WT(GDP) tubulin were collected at the ID23-1 beamline (European Synchrotron Radiation Facility, Grenoble, France), data for WT(GMPCPP) tubulin at the Proxima-1 beamline (SOLEIL Synchrotron, Saint-Aubin, France), and data for both Y222F(GDP) and (GTP) tubulin at Proxima-2 beamline (SOLEIL Synchrotron). They were processed with XDS (Kabsch, 2010). The sT$_2$R structure (PDB ID 3RYC) was used as a starting point for refinement with BUSTER (Bricogne et al., 2017) with iterative model building in Coot (Emsley et al., 2010). Data collection and refinement statistics are reported in Table S1. The atomic coordinates and structure factors have been deposited in the Protein Data Bank under accession numbers 6TIS (WT(GDP) tubulin), 6TIY (WT(GMPCPP) tubulin), 6TIZ (Y222F(GDP) tubulin), and 6TIU (Y222F(GTP) tubulin). Figures of the structural models were generated with PyMOL (www.pymol.org).

### Turbidimetry

Polymerization of MTs (in BRB80 buffer containing 1 mM GTP) was monitored by turbidimetry at 350 nm on a spectrofluorometer (FP-6500; JASCO). A micro quartz cell with an optical path of 3 mm (FMM-100; JASCO) was placed in a cell holder maintained at 25 ± 0.2°C. Polymerization was started by

introducing a 50-µl tubulin sample, kept at 0°C, into the cell. The temperature of the sample solution reached 25°C with a relaxation time of 4 s, monitored by a fluorescent temperature indicator (Kato et al., 1999). In both WT and Y222F, the turbidity returned to the original level when the temperature was lowered to 4°C.

## Darkfield microscopy observation of MTs

At a desired time point in the course of turbidity measurement, an aliquot of MT solution was sampled, diluted in BRB80 buffer solution containing 1 mM GTP for optimal observation of individual MT filaments, and introduced into a flow chamber prepared from a coverslip and a glass slide (no. 1 coverslip and FF-001 glass slide, respectively, from Matsunami Glass) that were spaced by double-sided Scotch tape (dimensions 9 mm × 9 mm × 80 µm). The chamber was sealed with VALAP, and the sample was observed under a darkfield microscope (BX50; Olympus) equipped with an objective lens (either Plan lens, 40×, NA = 0.65, or Plan lens, 20×, NA = 0.4; Olympus) at 25 ± 1°C. To avoid any influence of the glass on nucleation and polymerization (Roostalu et al., 2015), the MTs freely floating in solution were observed. The images were projected onto an image-intensified CCD camera (C7190; Hamamatsu Photonics) and stored in a digital video recorder (GV-HD700; Sony). The images were recorded between 30 s and 2 min after the dilution of MTs, during which time frames the number and length distribution of the MTs were virtually unchanged. To avoid aggregation of MTs, we did not use the cross-linking reagents to fix MTs, except in the case of the Y222F MTs shown in Fig. 1, O and P. In Fig. 1, O and P, to resolve the transient short MTs, the MTs were fixed with 1% glutaraldehyde, and the images of the MTs attached to the surface of the glass slide were recorded.

## Rapid flush negative stain EM

For WT (either GTP or GDP loaded) and Y222F(GDP), the reaction of polymerization was started by transferring the tubulin sample from a water bath set at ∼0°C to a water bath at 25°C. 1 min before the termination of the reaction, 50 µl of 1% uranyl acetate, 7 µl of air, and 0.5–3 µl of 10 µM WT or 5 M Y222F tubulin in BRB80 buffer were sequentially drawn into a pipette tip attached to a Gilson P200 Pipetman (Gilson) at 25°C. At the time point for the termination of the reaction (the exact numbers are indicated in the graph legend for Fig. 2 G), the entire contents of the tip were ejected onto a carbon film of an EM grid (Okenshoji). The rapid flush method allows immediate dilution and fixation of the protein samples within 30 ms (Frado and Craig, 1992; Imai et al., 2015). In the case of Y222F(GTP), the protocol was arranged to keep up with the rapid nucleation. The reaction of polymerization was started by drawing 1–2 µl of 5 µM Y222F(GTP) tubulin (in BRB80 buffer) at 4°C into a pipette tip at 25°C, which was preloaded with 7 µl of air and 50 µl of 1% uranyl acetate solution. 15 s later, the polymerization reaction was terminated by ejecting the entire contents of the tip onto the EM grid, as described above. In all experiments (WT and Y222F, either GTP or GDP loaded), after blotting off the excess stain, the grids were dried at room temperature. The samples were observed with a JEM1400Plus electron microscope (JEOL) operated at 80 kV. The images were recorded on an EM-14800RUBY CCD

camera (JEOL) at nominal 40,000 magnification (for the analysis of oligomers; Fig. S2, E–H) or nominal 2,500 magnification (for examination of MTs; Fig. S2, A–D), each corresponding to a pixel size of 0.413 or 6.59 nm, respectively.

## Measurement of oligomer curvature and length

For the analyses of curvature (Fig. 2, F and G), dimers and tetramers (i.e., the oligomers with a length <23 nm) were excluded from the analyses because they are too short for accurate curvature measurement. To avoid the overlap and/or superimposition of the oligomers, the measurement was made at oligomer densities threefold to fivefold lower than the densities shown in Fig. 2 E. For each oligomer, the arc length and radius of curvature were manually measured by drawing a polyline along the central trajectory of the oligomers and fitting a circle from the polyline by using ImageJ (https://imagej.nih.gov/ij/; Fig. 2 D). From the radius of curvature $r$ (nm), the curvature $\kappa$ (µm$^{-1}$) and the kink between the tubulin dimer $\theta$ (degree) were calculated by using the relationships $\kappa = 1,000/r$ and $\theta = 360 * (l/2\pi r)$, with an average dimer length, $l$, of 8 nm (Fig. 2, F and G). The measurement was performed by two independent researchers who were blinded to sample identification.

For quantitation of the length distribution of oligomers (Fig. 4 D), the length of all oligomers in 10–12 micrographs (with size 843 × 847 nm), each covering ∼1,000 oligomers in total, were manually measured using ImageJ (the exact numbers are indicated in the legend of Fig. 4). The experiments were performed twice for each condition (10 µM WT(GTP) at 5 min, 5 µM Y222F(GTP) at 15 s, and 5 µM Y222F(GDP) at 30 min after the onset of the polymerization reaction). For the oligomers with a length >12 nm, the central trajectory of the oligomer was fit by a circle, and the arc length was measured. For the oligomers with a length <12 nm, the largest distance between two points within the object (Ferret diameter) was measured.

For measurement of the length of each strand in the two-stranded oligomer complex (Fig. 6), the central trajectory of each strand was traced by drawing a polyline using ImageJ, and the total length of polyline was calculated. If the complex was twisted and two strands were partially superimposed (for example, Fig. 6, E and F), we measured only the part of a strand where individual strands could be identified.

## In vitro MT dynamics assays

The surface of the glass slides (FF-001; Matsunami Glass) was coated with biotin-functionalized polyethylene glycol (PEG; Biotin-PEG-SC molecular weight = 3,400; Laysan Bio; Bieling et al., 2010). A microscope chamber (18 mm × 6 mm × ∼100 µm) was constructed using a coverslip (no. 1 coverslip; Matsunami Glass) and a PEG-coated glass slide. GMPCPP-stabilized and biotinylated MT seeds were prepared from WT tubulin as described in Gell et al., 2010. For a dynamic assay, the GMPCPP seeds were immobilized on the bottom of the chamber (PEG-coated glass) via streptavidin (S888; Thermo Fisher Scientific), and 20 µl of WT or Y222F tubulin in BRB80 buffer solution containing 1 mM GTP was infused into the chamber. The chamber was sealed by vacuum grease and nail enamel, and the sample was observed under a darkfield microscope (BX50; Olympus; Plan lens, 20×,

NA = 0.4; Olympus) at 25 ± 1°C. Time-lapse images of the MTs were projected onto a high-sensitivity CCD camera (WAT-910HX/RC; Watec) at 1 frame/s, and the data were stored on a hard disc (DFG/USB2pro converter; The Imaging Source; Latitude 7370; Dell).

### Dynamic parameter measurement

Kymographs were generated from the time-lapse darkfield images of MTs using ImageJ. The MT growth rate was determined from the slope of the growing MT in the kymograph (Fig. 3, E–I). The mean and SD of growth rate were calculated from all MTs analyzed for each condition (the number of MTs analyzed per data point was 4–44; a total of 217, 122, and 235 MTs were analyzed for WT(GTP), Y222F(GTP), and Y222F(GDP) datasets, respectively). From the tubulin concentration dependence of the MT growth rate, $v_0[/s] = k_+C_0 - k_-$, the rate constants for the association and dissociation of tubulin to the MT plus end ($k_+$ and $k_-$) were calculated assuming that a MT with a length of 1 μm corresponds to 1,634 dimers (Walker et al., 1988; Fig. 3, I and K). Neither WT(GTP) nor Y222F(GTP) tubulin grew onto the minus end, whereas some minus-end elongation was observed with Y222F(GDP) tubulin. The catastrophe frequency was determined as the number of observed catastrophe events divided by the total time spent in the growth phase (363–2,491, 112–1,596, and 415–576 min per data point for WT(GTP), Y222F(GTP), and Y222F(GDP) tubulin, respectively). The SD was estimated by the catastrophe frequency divided by the square root of the number of catastrophes, assuming the catastrophe events are Poisson processes (Fig. 3, I and K). Assuming that the catastrophe frequency declines linearly with the tubulin concentration, we estimated the range of tubulin concentration where the catastrophe is suppressed (x intercept; Fig. 3 K).

### Assessment of the size of critical nucleus

Assuming that the turbidity is proportional to the amount of tubulin polymerized into MTs, the turbidity (Fig. 1, I and J) was converted to the concentration of tubulin in MTs (Fig. 4, A and B) by dividing the OD$_{350}$ values by the turbidity coefficient (slope of the lines in Fig. 1 K; Mirigian et al., 2013), 9.14 and 13.67 for WT and Y222F, respectively. The linearity between the plateau value of turbidity and the initial tubulin concentration justifies this conversion (Fig. 1 K).

We applied the standard nucleation-and-growth model (Oosawa and Asakura, 1975) to these kinetic curves of nucleation. The oligomers of critical size are, by definition, the minimal assemblies that are destined to grow. As entering into this stage is a very rare event, we supposed that the oligomers not exceeding the critical size are almost in equilibrium (quasi-equilibrium), and are undergoing the stochastic growth and degradation. In that condition, the nucleation rate, $I_0$ (the rate of increase in the number concentration of critical nucleus), is given by

$$I_0 = B(C_0)^\alpha, \qquad (1)$$

where $C_0$ is the concentration of the tubulin dimers in solution, $\alpha$ represents the size of the critical nucleus, and $B$ is a constant.

In the regime where the turbidity is <10% of the plateau value ($t < T_{10\%}$), $C_0$ can be regarded as constant. The constant $B$ reflects the specificities of the formation and configuration of the critical nucleus. In cases where the critical nucleus is formed by some conformational change of a linear oligomer of size $\alpha$, $B$ would be the equilibrium constant of the assembly of $\alpha$ dimers multiplied by the rate constant of the conformational change. However, in the case where irreversible growth is triggered by the incorporation of the $\alpha$-th tubulin dimer into the least stable oligomer, $B$ would reflect the binding rate of the last dimer in a specific conformation. In both cases, $B$ reflects the lateral interaction in some manner, and we have the above form for $I_0$.

While we know that the single-stranded oligomers and MTs coexist in the solution (Fig. 2, A–C; and Fig. S2), we have little information regarding the reaction cascade between the formation of critical nucleus and the appearance of a tubular form of MT. This passage may take some time, which we tentatively denote by Δ (see below).

Once the tubular form of the MT is achieved, the MT elongates at an average growth rate of $v_0$, which can be separately measured in an MT dynamics assay (Fig. 3 I). Then the nucleation events taking place in the interval time τ and τ + $d\tau$ have the concentration $I_0 d\tau$, and each of those events contributes to the MT mass through the form $v_0 \times (t - \tau - \Delta)$ at time $t$ (>τ +Δ). In the early phase of polymerization ($t < T_{10\%}$), the concentration of tubulin is high enough that we can ignore the catastrophe (Fig. 3, J and K). Therefore, the total mass of MTs at time $t$ is

$$\text{MT(t)} = I_0 \int_0^{t-\Delta} v_0(t - \tau - \Delta)d\tau = \frac{1}{2}I_0 v_0(t - \Delta)^2. \qquad (2)$$

This equation predicts that in the early stage of polymerization, the polymerization curve rises quadratically with time with a quadratic coefficient of $I_0 v_0 / 2$. Indeed, polymerization kinetics from time zero to $T_{10\%}$ fits very well to the quadratic function (Fig. 4, A and B). By substituting the growth rate $v_0$ measured in an MT dynamic assay (Fig. 3, E–K), we can find $I_0$ at each concentration $C_0$. In Fig. 4 C, the log–log plot of $I_0$ vs. $C_0$ gives a line with a slope corresponding to the size of critical nucleus (α), 4.0 ± 0.2 and 5.9 ± 0.3 for WT (blue filled circle) and Y222F (red filled circle) tubulin, respectively. We checked how the inclusion of the lag time Δ in Eq. 2 affects α. While the introduction of the lag Δ in Eq. 2 improves the quadratic fit (R improved from 0.98 to 0.99 for both WT and Y222F), it affects the value of α by no more than 7%. These results justify our estimation.

In estimating the size of critical nucleus, we were concerned about the possible errors due to the nonlinearity of OD$_{350}$ to the polymer mass of very short MTs. In the above calculation for Y222F tubulin, the data measured for 4 and 5 μM tubulin were not included, because some of the MTs could be too short to assure the linearity between the turbidity and the mass of MTs (Berne, 1974). When we included the data for 4 and 5 μM tubulin in the fitting (Fig. 4 C, red filled and open circles), α was 5.4 ± 0.2, virtually the same as the calculation excluding these two datasets.

The size α calculated by an alternative method also confirms the validity of our method. The assessment of the size from the power law dependence of $1/T_{10\%}$ on tubulin concentration

(Fig. 1 L; Voter and Erickson, 1984) showed α for WT and Y222F tubulin to be 3.5 ± 0.1 and 6.1 ± 0.4, respectively, close to the values obtained by our method (4.0 ± 0.2 and 5.9 ± 0.3 for WT and Y222F tubulin, respectively). The calculation based on the 10% criterion ($1/T_{10\%}$) slightly underestimates the size α for WT tubulin because at the tubulin concentrations near the critical concentration (for example, at 10 µM tubulin), $T_{10\%}$ is affected not only by the nucleation and elongation but also by the catastrophe, breaking the premise for this calculation that $1/T_{10\%}$ is determined only by the nucleation and elongation (Oosawa and Asakura, 1975).

The size of the critical nucleus estimated for Y222F tubulin is comparable to the size estimated for brain tubulin where the nucleation was accelerated by glycerol (α = 7.5), calculated by the power law dependence of $1/T_{10\%}$ on tubulin concentration (Voter and Erickson, 1984, according to their definition of critical nucleus, α = 6.0).

### Assessment of the size of critical nucleus for templated nucleation

Zheng et al. (1995) counted the number of MTs, $N_{MT}$, formed in the initial 5 min of incubation of brain tubulin at 37°C both in the presence and absence of γTuRC (Fig. 3 in Zheng et al., 1995). The dependence of this number on the tubulin concentration is fit by the equation

$$N_{MT} \sim B(C_0)^{\alpha}. \quad (3)$$

The size of the critical nucleus, α, for the nucleation templated by γTuRC was then estimated as 6.7 ± 0.9 (Fig. S4 B).

### Calculation of the molar concentration of 2n-mer, $x_{2n}$

Based on the dataset for oligomer length, we first plotted a cumulative frequency of length (Fig. 5, A–C). These plots were then numerically converted into an empirical probability density as a function of oligomer length. After smoothening the densities using a Gaussian filter, we read off each threshold length separating the first and second, the second and third, and the third and fourth populations, respectively (i.e., 2n-mer and 2[n+1]-mer with n = 1–3). For those thresholds between the oligomers of larger size (n > 3), they were determined by extrapolating the average spacing between the thresholds for the smaller sizes (n ≤ 3). The probability of 2n-mer, $p_{2n}$, was then calculated by integrating the probability density over the interval between neighboring threshold lengths.

The next step is to connect $p_{2n}$ to the molecular concentrations ($x_{2n}$). At the time point when the polymerization reaction was terminated on the EM grids, the turbidity reached 7.0% (WT(GTP)) and 25.3% (Y222F(GTP)) of the plateau value. Considering the initial tubulin concentration ($C_0$; 10 and 5 µM for WT(GTP) and Y222F(GTP) tubulin, respectively) and the critical concentration ($C_c$; 4.7 and 0.19 µM for WT(GTP) and Y222F(GTP) tubulin, respectively), the concentration of tubulin dimers contained in the ensemble of oligomers ($\sum_{n=1}^{\infty} nx_{2n}$) is estimated to be 9.6 and 3.8 µM for WT(GTP) and Y222F(GTP) tubulin, respectively. For Y222F(GDP) tubulin, $\sum_{n=1}^{\infty} nx_{2n}$ is estimated to be 4.9 µM, taking the polymerized 0.1 µM tubulin (estimated from the turbidity at the time the reaction was terminated) into

account. Finally, the molar concentration of each 2n-mer ($x_{2n}$) was calculated from the total concentration of oligomer, $\sum_{n=1}^{\infty} nx_{2n}$, and the probability of each 2n-mer, $p_{2n}$ (Fig. 5, D–F). Because the calculation neglects the small population of multi-stranded complexes or sheets that are too small to contribute the turbidity (Berne, 1974), the total concentration of oligomers might include some errors, giving only an upper limit for $\sum_{n=1}^{\infty} nx_{2n}$.

### Calculation of the specific free energy gains, $\Delta G_{olig}$ and $\Delta G_{MT}$, upon binding of tubulin dimer

Assuming that the ensemble of prenucleus oligomers with various sizes is in rapid quasi-equilibrium, the free energy gain of an oligomer upon tubulin binding ($\Delta G_{olig}$) was calculated from the concentrations of oligomers with different sizes (Fig. 5, D–F). The condition of chemical equilibrium reads

$$\mu_{2(n-1)} + \mu_2 = \mu_{2n}, \quad (4)$$

where

$$\mu_{2n} = \mu_{2n}^0 + k_B T \ln x_{2n} \quad (5)$$

is the chemical potential of 2n-mer, with $\mu_{2n}^0$ representing the concentration-independent part. By substituting Eq. 5 into Eq. 4, the free energy change upon the assembly of 2n-mer from 2(n-1)-mer can be represented by

$$\left[\Delta G_{olig}\right]_{2(n-1)}^{2n} = \mu_{2n}^0 - \mu_{2(n-1)}^0 - \mu_2^0 = -k_B T \ln \frac{x_{2n}}{x_{2(n-1)}x_2}. \quad (6)$$

Based on the concentrations of oligomers with various sizes $x_2$, $x_4$, $x_6$... (Fig. 5, D–F), the values $\left[\Delta G_{olig}\right]_{2(n-1)}^{2n}$ were obtained (Fig. 5 G). For WT(GTP) and Y222F(GDP) tubulin, exponential decay of $x_{2n}$ with n indicates that $\Delta G_{olig}$ is constant independent of the size of oligomer (Fig. 5, D and F). Thus, $\Delta G_{olig}$ was calculated from the slope of the linear fitting in Fig. 5, D and F, and plotted in Fig. 5 G, with each error bar representing the error of the fit. For Y222F(GTP) tubulin, $\Delta G_{olig}$ for each 2n-mer was calculated as the local slope of the quadratic fitting in Fig. 5 E and plotted in Fig. 5 G (no error bars).

In analogy with Eq. 6, when a MT containing 2(N-1) tubulin dimer incorporates another dimer, the free energy change upon binding should be obtained from the ratio $\frac{MT_{2N}}{MT_{2(N-1)}C_0}$ as

$$\left[\Delta G_{MT}\right]_{2(N-1)}^{2N} = -k_B T \ln \frac{MT_{2N}}{MT_{2(N-1)}C_0}, \quad (7)$$

where $C_0$ is the concentration of tubulin dimer, where we have ignored the consummation of dimer in oligomers. On the other hand, the binding equilibrium is determined by the rate constant for association and dissociation of tubulin dimer ($k_+$ and $k_-$) at the growing end of an MT,

$$\frac{MT_{2N}}{MT_{2(N-1)}C_0} = \frac{k_+}{k_-}. \quad (8)$$

The rate constants, $k_+$ and $k_-$, can be calculated from the concentration dependence of the growth rate of MT, $v_0 = k_+ C_0 - k_-$ (number of tubulin dimers added onto each MT filament per second), measured by in vitro MT dynamic assay (Fig. 3, I and K). Thus, the free energy gain upon binding of a tubulin dimer to

an MT was calculated by the following equation and shown in Fig. 5 G:

$$\left[\Delta G_{\text{MT}}\right]_{2(N-1)}^{2N} = -k_{\text{B}}T \, ln\frac{k_+}{k_-}. \qquad (9)$$

The error bars for each $\Delta G_{\text{MT}}$ in Fig. 5 G represent the errors estimated from the SD of $\frac{k_+}{k_-}$. It is worth noting that while the energy changes $\left[\Delta G_{\text{olig}}\right]_{2(n-1)}^{2n}$ and $\left[\Delta G_{\text{MT}}\right]_{2(N-1)}^{2N}$ can contain an additive correction related to the choice of the standard volume for the mixing entropy calculation, the difference between these two energies, which should reflect the free energy change associated with the lateral interaction between tubulin dimers, does not suffer from it. The difference gives the absolute meaning.

### Statistical methods
Data analyses were conducted in Microsoft Excel 2013 (Microsoft Office), Kaleidagraph 4.5 (Hulinks) and ad hoc programs written by Mathematica 8.0.4. (Wolfram Research).

### Data availability
The atomic coordinates and structure factors have been deposited in the Protein Data Bank under accession numbers 6TIS, 6TIY, 6TIZ, and 6TIU. The data that support the findings of this study are available from the corresponding authors upon reasonable request.

### Online supplemental material
Fig. S1 shows preparation of *Drosophila* tubulin and cryoEM images of WT and Y222F MTs. Fig. S2 shows the negative stain EM images of WT and Y222F MTs and oligomers recorded in the course of polymerization. Fig. S3 shows EM images of WT and Y222F tubulin samples used for turbidimetry (4°C). Fig. S4 shows an analysis of the size of the critical nucleus for the case of γTuRC-dependent nucleation. Table S1 reports the x-ray data collection and refinement statistics. Table S2 shows the conservation of the amino acid residue Y222 in the various species across kingdoms.

## Acknowledgments

We acknowledge the Chuo University Electron Microscopy Facility and technical assistance by Ms. Nakahara for the negative stain EM. Part of the cryoEM study was conducted at the Advanced Characterization Nanotechnology Platform of the University of Tokyo, supported by the Nanotechnology Platform of the Ministry of Education, Culture, Sports, Science, and Technology of Japan. Diffraction data were collected at the SOLEIL synchrotron (PX1 and PX2 beam lines, Saint-Aubin, France) and the European Synchrotron Radiation Facility (ID23-1 beam line, Grenoble, France). We are most grateful to the machine and beam line groups for making these experiments possible. Special thanks are also due to Dr. Masayuki Kajitani for providing tobacco mosaic viruses, the Drosophila Genomics Resource Center (supported by National Institutes of Health grant 2P40OD010949) for providing *Drosophila* tubulin genes, Dr. Yuichiro Maeda and Dr. Takahide Kon for critical reading of the manuscript, Dr. Hiroko Takazaki and Ms. Yoshimi Asano for assistance in sample preparation, Dr. Valérie Campanacci for image analysis, and Dr. Marcel Knossow and the late Fumio Oosawa for their discussion and continuous encouragement.

This work was supported by the Grants-in-Aid for Scientific Research from the Ministry of Education, Culture, Sports, Science, and Technology of Japan (17H03668 to E. Muto, S. Kamimura, and H. Imai). Additional support for this work came from the Fondation ARC pour la Recherche sur le Cancer (PJA20161204544 to B. Gigant).

The authors declare no competing financial interests.

Author contributions: I. Minoura, E. Muto, and B. Gigant designed the Y222F mutant and the experiments. S. Uchimura and R. Ayukawa developed the experimental system for the expression and purification of *Drosophila* tubulin. S. Iwata, H. Imai, R. Ayukawa, and S. Kamimura performed negative stain EM. S. Iwata, H. Imai, M. Shirouzu, and H. Shigematsu performed the cryoEM work, and S. Iwata, R. Ayukawa, K.X. Ngo, M. Hayashi, and B. Gigant conducted oligomer image analysis. B. Gigant and T. Makino conducted crystal structure analysis. R. Ayukawa, E. Muto, and K. Sekimoto conducted turbidimetry and kinetic analysis, and M. Hayashi measured the in vitro MT dynamics. E. Muto and K. Sekimoto conducted statistical analyses of oligomers and built a model. E. Muto, K. Sekimoto, and B. Gigant prepared the manuscript. All co-authors discussed the results and commented on the manuscript.

Submitted: 10 July 2020

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

# Supplemental material

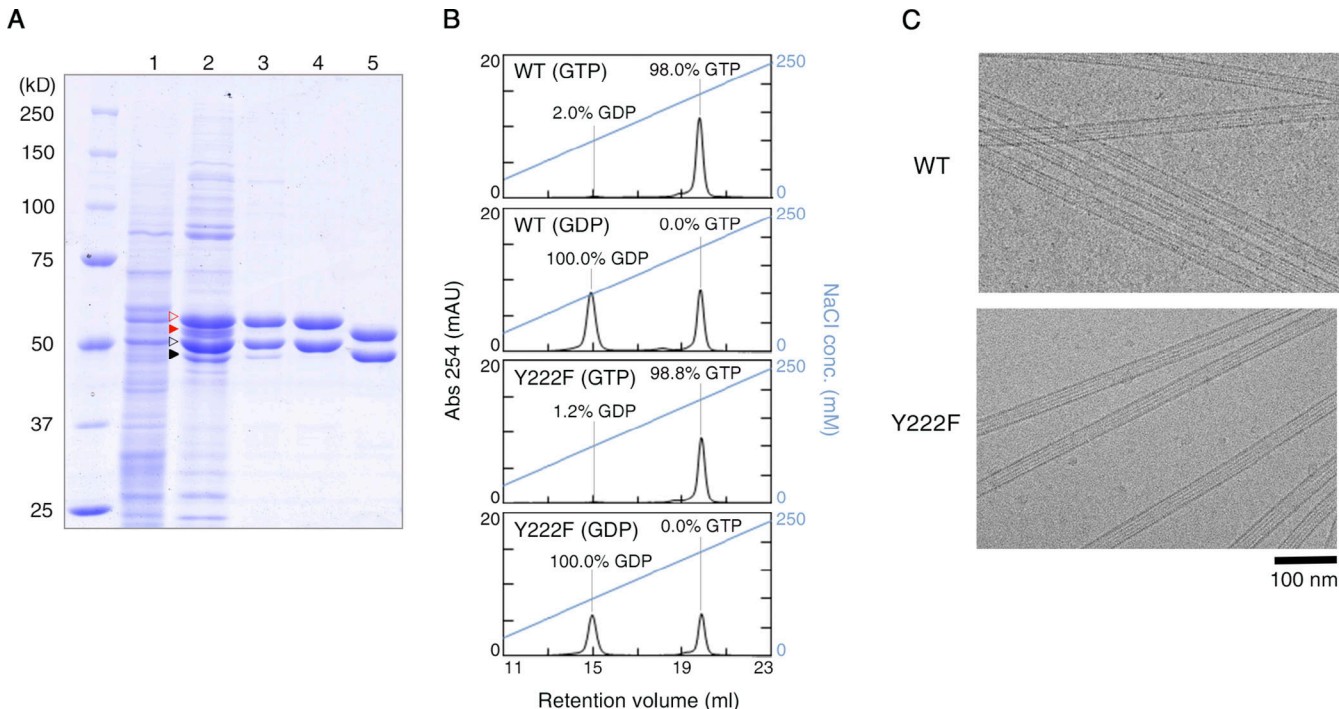

Figure S1. **Preparation of *Drosophila* tubulin. (A)** SDS PAGE showing each step in preparation of WT tubulin. Lanes: (1) Cell lysate. (2) Diethylaminoethyl sepharose anion exchange column eluent. Red and black open arrowheads, recombinant α- and β-tubulin; red and black filled arrowheads, endogenous α- and β-tubulin, respectively. (3) His-tag affinity column eluent. (4) FLAG-tag affinity column eluent. (5) Final product after tag cleavage. **(B)** Analysis of nucleotide contents by ion exchange chromatography. The percentage indicates the occupancies at the E-site calculated from each chromatogram, assuming equal numbers of exchangeable and nonexchangeable nucleotide sites per tubulin dimer and that the nonexchangeable nucleotide is GTP. mAU, milli-absorbance unit. **(C)** CryoEM images of WT and Y222F MTs.

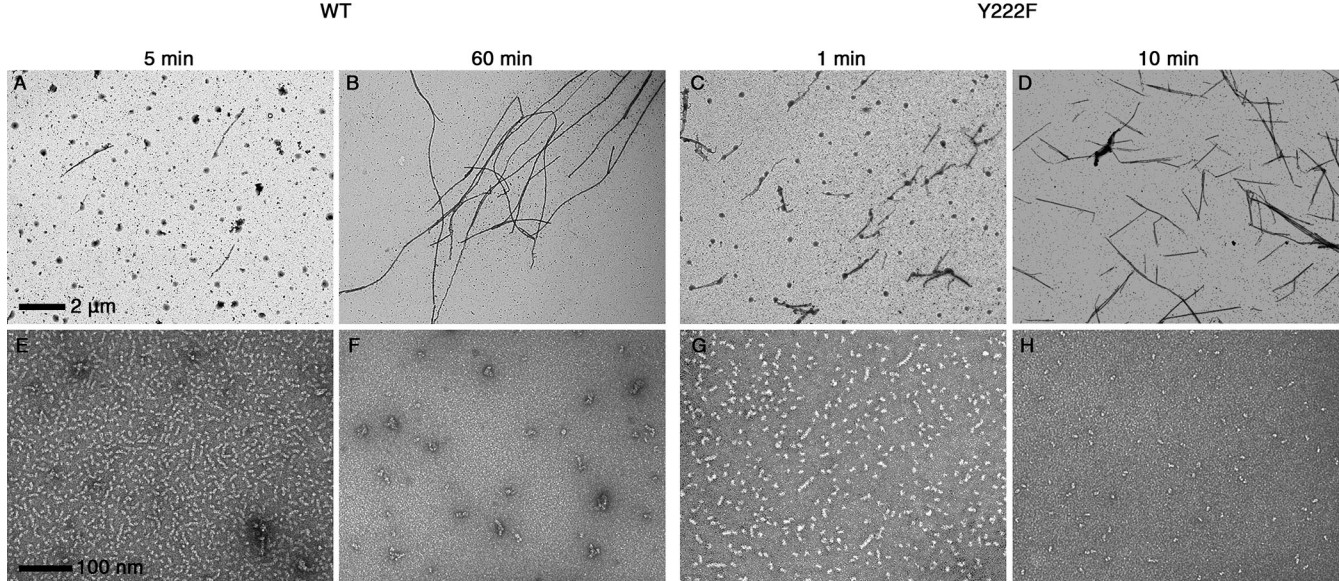

Figure S2. **The number of oligomers decreased with the progress of MT assembly.** EM images of WT MTs (A and B) and oligomers (E and F), Y222F MTs (C and D) and oligomers (G and H) recorded in the course of polymerization (Fig. 1, I and J). The initial concentrations of tubulin were 10 μM (WT) and 5 μM (Y222F). The photographs of MTs and oligomers were taken at 2,500× and 40,000× nominal magnification, respectively. For Y222F mutant tubulin, only the dimers and aggregates are seen at 10 min, which is when the turbidity reached ~90% of the plateau value (H). In contrast, for WT tubulin, the oligomers and aggregates existed at 60 min, which is when the turbidity reached a plateau (F).

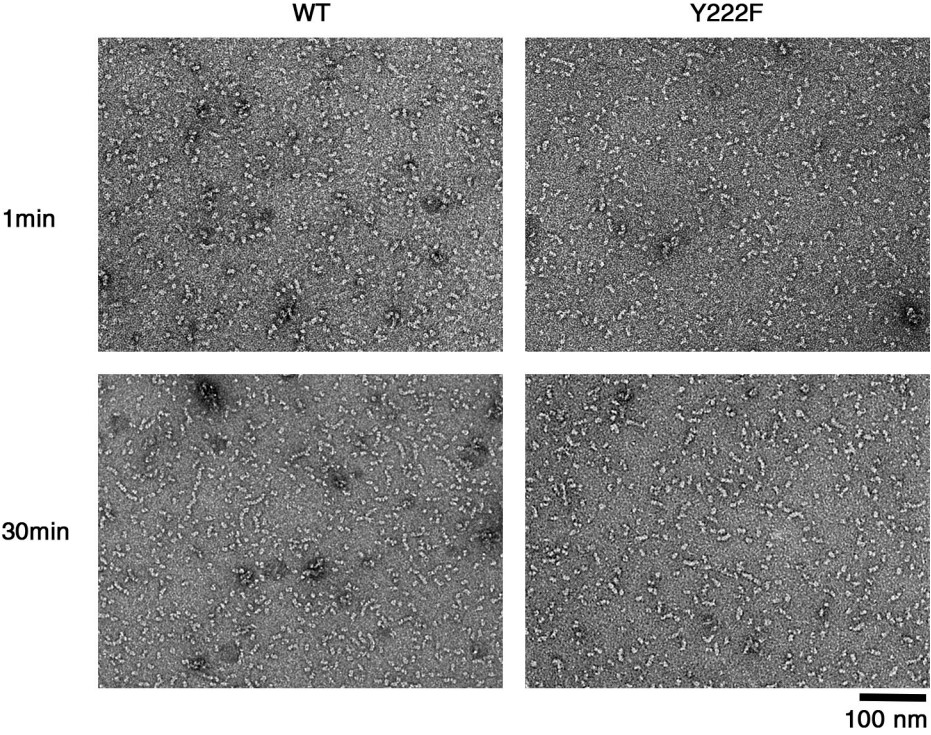

Figure S3. **EM images of the tubulin samples used for turbidimetry.** EM images of WT and Y222F tubulin 1 min after the ultracentrifugation (top) and 30 min later (bottom) at 4°C. Although our ultracentrifugation condition (see Materials and methods) should sediment oligomers that are larger than dimers, we sometimes observed a tetramer in the Y222F sample immediately after ultracentrifugation, which may have been assembled during the 1 min needed for the preparation of the EM grid.

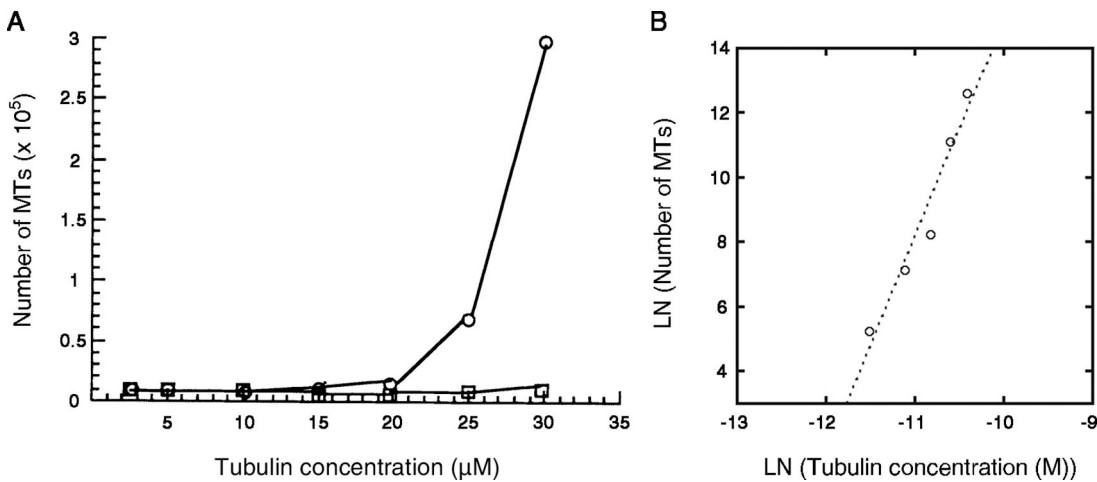

Figure S4. **Initiation of MTs by the γTuRC template. (A)** Data from Fig. 3a in Zheng et al., 1995. The number of MTs newly formed within 5 min of incubation at 37°C with and without γTuRC (circle and square, respectively) at variable tubulin concentrations. **(B)** Dataset for γTuRC-dependent nucleation plotted on a log–log scale. The data can be fit by the equation y = 81.5 + 6.7x (R = 0.97), indicating that the nucleation of MTs requires the formation of a critical nucleus composed of approximately seven tubulin dimers. We could not estimate the size of the critical nucleus for spontaneous nucleation (without γTuRC) because MTs were barely formed. Fig. S4 A is reprinted with permission from *Nature*.

**Table S1 and Table S2 are provided online as separate files. Table S1 shows data collection and refinement statistics. Table S2 shows the nature of the β-tubulin residue 222 in different organisms.**

