## [Peer Review File · The Journal of Cell Biology]

GTP-dependent formation of straight tubulin oligomers leads to microtubule nucleation

Rie Ayukawa, Seigo Iwata, Hiroshi Imai, Shinji Kamimura, Masahito Hayashi, Kien Ngo, Itsushi Minoura, Seiichi Uchimura, Tsukasa Makino, Mikako Shirouzu, Hideki Shigematsu, Ken Sekimoto, Benoît Gigant, and Etsuko Muto

Corresponding Author(s): Etsuko Muto, RIKEN Center for Brain Science

Review Timeline:

Submission Date:	2020-07-10
Editorial Decision:	2020-08-07
Revision Received:	2020-11-23
Editorial Decision:	2020-12-10
Revision Received:	2020-12-28

Monitoring Editor: Arshad Desai

Scientific Editor: Melina Casadio

Transaction Report:

DOI: <https://doi.org/10.1083/jcb.202007033>

August 7, 2020

Re: JCB manuscript #202007033

Dr. Etsuko Muto
RIKEN Center for Brain Science
Hirosawa 2-1
Wako, Saitama 351-0198
Japan

Dear Dr. Muto,

Thank you for submitting your manuscript entitled "GTP-dependent formation of straight oligomers leads to nucleation of microtubules" and thank you for your patience with the peer review process. The manuscript has been evaluated by 3 expert reviewers, whose feedback is appended to this letter. Based on their evaluations, we are interested in receiving a revised version of your manuscript.

All three reviewers find your analysis of the Y222F mutant compelling but they raise a number of points that will need to be addressed in the revision. In particular, please address major points on the mechanistic interpretation of the Y222F mutation (Points #1 from Rev #2 & from Rev #3), potentially through consideration of alternative mechanisms to the one you propose and/or by including experimental data that may help address their comments (we note that an alternative mechanism may not need to be explicitly detailed; it is perfectly acceptable to state that the effect of the mutant may be due to reasons that require future explication). Rev #2 also raises the possibility that the Y222F mutation exhibits more nucleation because it enhances oligomer formation - potentially this point is addressable from data comparing the wild-type and mutant tubulins at identical concentrations. Please also address points related to curvature measurement (Revs #1 & #2) and follow suggestions on text and figures that are aimed at improving the accessibility of the manuscript. In our view, all of the reviewer points seem valid and are aimed at key issues on interpretation of the mutant tubulin and improving the manuscript during revision. We note that we are not requiring significant new additional experimentation; rather, we encourage you to use existing data and/or to modify the text & figures in light of the reviewer feedback. Please also include a response to all of the reviewer points, highlighting changes made in the manuscript, when submitting your revision.

GENERAL GUIDELINES:

Text limits: Character count for an Article is < 40,000, not including spaces. Count includes title page, abstract, introduction, results, discussion, acknowledgments, and figure legends. Count does not include materials and methods, references, tables, or supplemental legends.

Figures: Articles may have up to 10 main text figures. Figures must be prepared according to the

policies outlined in our Instructions to Authors, under Data Presentation, <http://jcb.rupress.org/site/misc/ifora.xhtml>. All figures in accepted manuscripts will be screened prior to publication.

*****IMPORTANT:** It is JCB policy that if requested, original data images must be made available. Failure to provide original images upon request will result in unavoidable delays in publication. Please ensure that you have access to all original microscopy and blot data images before submitting your revision.*******

Supplemental information: There are strict limits on the allowable amount of supplemental data. Articles may have up to 5 supplemental figures. Up to 10 supplemental videos or flash animations are allowed. A summary of all supplemental material should appear at the end of the Materials and methods section.

As you may know, the typical timeframe for revisions is three to four months. However, we at JCB realize that the implementation of social distancing and shelter in place measures that limit spread of COVID-19 also pose challenges to scientific researchers. Lab closures especially are preventing scientists from conducting experiments to further their research. Therefore, JCB has waived the revision time limit. We recommend that you reach out to the editors once your lab has reopened to decide on an appropriate time frame for resubmission. Please note that papers are generally considered through only one revision cycle, so any revised manuscript will likely be either accepted or rejected.

Thank you for this interesting contribution to Journal of Cell Biology. You can contact us at the journal office with any questions, cellbio@rockefeller.edu or call (212) 327-8588.

Sincerely,

Arshad Desai, PhD
Editor, Journal of Cell Biology

Melina Casadio, PhD
Senior Scientific Editor, Journal of Cell Biology

Reviewer #1 (Comments to the Authors (Required)):

In the paper "GTP-dependent formation of straight oligomers leads to nucleation of microtubules", the authors examine the role of oligomer curvature in microtubule nucleation. By using a mutation that increases the proportion of straight oligomers, the authors find that nucleation is accelerated by the presence of straight oligomers. The authors make the interesting argument that straight oligomers are compatible with lateral association of new tubulin subunits or other oligomers, which then allows the growth of microtubules.

Overall, this is a well written, interesting paper, with strong data and analysis. The conclusions are carefully written and seem well supported by the data. Thus, I only have minor comments and

suggestions, as follows:

- 1) Fig. 3d: how is curvature measured on very short oligomers/dimers? Are short oligomers/dimers eliminated from the analysis? There is no clear difference between WT and mutant in the example images shown - it would perhaps be more useful to show WT-GDP vs Mutant-GDP, which has the biggest contrast? Or all 4 conditions? This image seems critical to demonstrate the quality of the curvature data.
- 2) The graphs in Fig. 3f-l are difficult to read generally, and nearly impossible to compare for each condition side-by-side. It may be useful to consider other graphical formats to allow for a clearer and more direct comparison between conditions. One idea may be to show curvature as a color change? Or show Dot plots with all points, length vs curvature?
- 3) Fig. 6 indicates that microtubules nucleate and grow starting with a single-stranded oligomer that reaches the size of the "critical nucleus", then laterally associates with a dimer or another oligomer, and then continues to grow as a multistranded oligomer that ultimately becomes a microtubule. This is an interesting result, but it is shown only in the mutant tubulin. Is this the normal process of nucleation and growth? IF so, can it be shown in WT tubulin, by using higher concentrations of free tubulin than 10 μ M?

Reviewer #2 (Comments to the Authors (Required)):

The authors investigate here the mechanism of microtubule nucleation in solution with purified recombinant *Drosophila* tubulin. They compare wild type tubulin and a single point mutant that is intended to alter the conformation of a loop at the surface of beta-tubulin in a nucleotide-dependent manner. This loop was previously proposed to mediate longitudinal tubulin interactions. Using turbidity measurements and darkfield microscopy, the authors find that this Y222F mutation does indeed promote microtubule nucleation and growth. They then study tubulin oligomers of wildtype and mutated tubulin in the presence of both GTP and GDP by negative stain electron microscopy and find a negative correlation between oligomer curvature and nucleation efficiency. Comparing oligomer sizes (from EM data) and critical nucleus sizes (from kinetic data) the authors conclude that the critical nucleus required for persistent microtubule growth is formed by the lateral association of 2 straight tubulin oligomers and that oligomer straightening promotes microtubule nucleation by promoting lateral association of oligomers. These are carefully performed experiments, providing new information regarding the long-standing question of the mechanism of microtubule nucleation. However, not all conclusions regarding the mechanism of nucleation appear to be compelling as stated at this stage of the manuscript.

Main points of criticism:

1. Mechanism by which the Y222F mutation affects nucleation: The Y222F mutation is expected to break an interaction with the T5 loop of beta-tubulin with GDP bound. According to previous data, in wild-type tubulin GTP induces a conformational change naturally breaking the loop interaction, moving the loop 'out' and consequently promoting longitudinal tubulin interactions. Considering this view, the main effect of the mutation should be expected for GDP tubulin. And indeed, the authors find that GDP-Y222F tubulin can polymerize remarkably well - in contrast to wild type tubulin. However, also GTP-Y222F tubulin nucleates and polymerizes better than GTP-wild type tubulin. This does not seem to agree with the view of the role of the T5 loop that the authors present here.

For both GTP tubulins, the loop should be 'out' and longitudinal tubulin interactions should be the same/similar for wild type and mutant in the GTP state. It seems unlikely that GTP is hydrolysed at the ends of oligomers (if it was, it should be substantial given the assumed transient nature of oligomers and could be measured). If oligomers do not hydrolyse GTP in significant amounts, then it seems that the mutation does something else to promote tubulin interactions in the GTP state. What is it?

2. Effect of oligomer concentrations: The authors focus here on differences in curvature of wild type and Y222F tubulin oligomers and argue that the observed curvature differences cause differences in the nucleation efficiency of different tubulins. If indeed the kinetic pathway to microtubule formation includes the lateral association of oligomers, one expects that the simply concentration of oligomers has a strong effect on microtubule nucleation. The authors observe tubulin oligomers for the Y222F mutant at lower tubulin concentrations than for wild type and in both cases seem to have a convenient density for their morphological analysis by EM. (The authors also estimate oligomer concentrations in Suppl. Fig. 5, but it was unclear if these oligomer concentrations referred to different tubulin concentrations for wild type and mutant.) Does this mean that the Y222F oligomer concentration per tubulin concentration is much higher than for wild type? What are the relative concentrations of oligomers per tubulin concentration for the 4 categories of wild type and Y222F tubulin with each GTP and GDP? Is possibly simply the much higher stability (and therefore concentration) of oligomers the main reason for the increased nucleation efficiency of Y222F tubulin, with the details of minor shape differences between different oligomer classes playing perhaps only a minor role?

Other points:

3. Scholarship: In the Introduction, it seems appropriate to clearly introduce previous published observations of tubulin oligomers and previous conclusions of their role for microtubule nucleation and to explain in the Discussion to which extent the observations here agree with previous work and to which extent this study goes beyond previous results and conclusions (e.g. Voter & Erickson, JBC, 1984; Wang et al., Cell Cycle, 2005; Mozziconacci et al., Plos One, 2008; Portran et al., NCB, 2017). In this context, the authors might want to rephrase their sentence in the Introduction that the visualization of nucleation intermediates is "impossible" if not using their method.

4. Fig. 2: Why is the structure of wild type tubulin presented with GMPCPP bound and the structure of the Y222F mutant with GTP bound? The authors could be clear about this in the main text. Can the nucleotide difference affect the conclusions drawn for the structure of these tubulins in the GTP state?

5. Fig. 3e-i: Why is the nucleotide-dependent difference between the distributions of oligomer curvature larger for wild type than for Y222F tubulin? How significant at all is the difference between GTP and GDP state for Y222F tubulin?

6. Same Fig.: Why are oligomers of the mutant in GTP straighter than those of the wild type in GTP?

7. Calculation of critical nucleus size (starting in line 195): Are oligomers assumed to grow only by plus end elongation? It seems that only plus end growth speeds have been used for the calculation of critical nucleus size. How does the size of the critical nucleus change if one assumes that oligomers can also elongate by minus end growth? Why is the critical nucleus for Y222F tubulin larger than for wild type? In other words, why does the nucleus for the tubulin that nucleates better

need to be bigger to be stable? This seems to be counter-intuitive.

8. Fig. 5d: Can the oligomer length distributions be related to the oligomer curvature distributions? Are oligomers straighter when they are longer? Taken together, it appears that the potentially nucleation-promoting characteristics of a reduced curvature and a longer length of Y222F oligomers may be compensated by the need for a larger nucleus to form. Maybe instead the larger critical nucleus is a consequence of longer oligomers (since they have to come together laterally to form a nucleus). Considering all these data together, it is a little unclear why Y222F tubulin nucleates better in the model of the authors.

9. Suppl. Fig. 4: Is the catastrophe frequency for the Y222F mutant reduced, because microtubules grow faster than wild type? A plot of the catastrophe frequency as a function of growth speed for the two tubulins would give a clear answer. Why are Y222F microtubules growing in GTP more stable than wild type microtubule in GTP?

In conclusion, the authors present here several very interesting correlations in the context of the nucleation efficiency of different tubulins in different nucleotide states. The model of the authors stating a set of causalities is plausible, but alternative models seem to be also possible. The mechanistic origin of the difference in nucleation efficiency between mutant and wildtype tubulin in the presence of GTP is probably not yet fully understood and could deserve some more discussion.

Reviewer #3 (Comments to the Authors (Required)):

This manuscript by Ayukawa et al. investigates a fundamental yet poorly understood question in cell biology: what are the limiting factors for microtubule nucleation? In this work, the characterizations of tubulin Y222F mutant provide tremendous insight into the first step of this process; and the use of rapid flash negative stain EM allows direct visualization of the elusive 'early nucleation intermediate' at unprecedented details. Overall, I think this manuscript represents an important breakthrough in understanding a key aspect of the microtubule biology. Therefore, I highly recommend publishing in JCB after addressing a few issues listed below.

1) In the polymerized state, as shown in the high-resolution cryo-EM structures of microtubule, the T5 loop of beta tubulin is in the 'in' conformation, regardless of the nucleotide state at the E-site (Zhang et al. PNAS 2018). This seems to be inconsistent with the major conclusion in this paper - the 'out' conformation of T5 is favored for microtubule nucleation/formation. However, it is highly possible that the T5 flips back "in" once the tubulins are incorporated into the lattice and/or upon GTP hydrolysis. I look forward to the authors' discussion regarding this issue.

2) The authors may consider merging or rearranging Fig. 1 and 2. The current panel Fig. 1a is a bit hard to follow for readers who are unfamiliar with tubulin structure, and it is rare to have a schematic picture as Fig. 1.

3) The authors wrote 'the negatively charged residue D177 in beta-tubulin is exposed towards the solvent, likely mediating the incoming tubulin dimer having positive charges on the interface (of alpha-tubulin) to establish a longitudinal contact'. Now that high-resolution structures are available for both curved and straight tubulin conformations, can the authors make an attempt to model the tubulin longitudinal interface for the early step of nucleation (i.e. straight oligomers), or at least be more specific about which positively charged residues of alpha tubulin are likely involved? Some of

the residues proposed in a previous paper (Nawrotek et al. JMB 2011) seem to be far away from D177 even in a 'out' conformation.

4) I am very curious what the 'early nucleation intermediates' of GMPCPP-microtubule look like, as visualized by rapid flash negative stain EM. Perhaps this event is too fast to be captured even by this technique. To be clear, I am not asking for new experiments, given the COVID19 situation, but the authors may consider including the GMPCPP data if they already have them in hand.

Minor issues:

5) The residue number Y222 of beta tubulin is inconsistent with the numbering Y224 in a previous paper by one of the co-authors (Nawrotek et al. JMB 2011). I understand this is due to alignment between alpha and beta tubulin, but it should be noted in the paper.

6) The current Sup Fig. 1 can be moved to the main figures after figure rearrangement.

Signed reviewer:

Rui Zhang

Washington University in St. Louis

Response to Reviewer #1

We thank this reviewer for her/his very positive evaluation of our work. Please find hereunder our response to this reviewer's comments.

Reviewer's comment

1) *Fig. 3d: how is curvature measured on very short oligomers/dimers? Are short oligomers/dimers eliminated from the analysis?*

The way the curvature has been measured was explained in the legend of Fig. 3d (new Fig. 2D). Only those oligomers comprising at least 3 heterodimers were subjected to the analyses. **Legend of Fig. 2**

There is no clear difference between WT and mutant in the example images shown - it would perhaps be more useful to show WT-GDP vs Mutant-GDP, which has the biggest contrast? Or all 4 conditions? This image seems critical to demonstrate the quality of the curvature data.

The electron micrographs presented in the previous Fig. 3d were not meant to show typical oligomer shapes but to help readers understand that the measurement was conducted at low oligomer density, where individual oligomers do not overlap with neighbors. However, we agree with the Reviewer's opinion that, here, we should provide representative images of oligomers supporting the quality of the curvature data. Thus, we replaced the electron micrographs by the images of oligomers at higher densities, with an annotation in legend explaining that the measurement was conducted at lower oligomer densities. **Fig. 2E**

2) *The graphs in Fig. 3f-I are difficult to read generally, and nearly impossible to compare for each condition side-by-side. It may be useful to consider other graphical formats to allow for a clearer and more direct comparison between conditions. One idea may be to show curvature as a color change? Or show Dot plots with all points, length vs curvature?*

We changed the graphs in Fig. 3f-i (now Fig. 2F) to dot plots, which may appear more familiar to the reviewer and (probably) to readers. **Fig. 2F**

What we wanted to show by the previous histograms is a two-dimensional probability density distribution, which is conceptually linked with our model (now Fig. 8). In the new dot plots, please note that the major populations are not responsible for nucleation. It is the marginal minor population that determines whether nucleation takes place (the area highlighted in yellow).

3) *Fig. 6 indicates that microtubules nucleate and grow starting with a single-stranded oligomer that reaches the size of the "critical nucleus", then laterally associates with a dimer or another oligomer, and then continues to grow as a multistranded oligomer that ultimately becomes a microtubule. This is an interesting result, but it is shown only in the mutant tubulin. Is this the normal process of nucleation and growth?*

We could not find WT multi-stranded oligomers probably because of the low nucleation rate. However, it is reasonable to expect the WT critical nuclei to form multi-stranded complexes because for the structural intermediates to overcome the energy barrier for nucleation, the dimension of the growth has to be shifted from one- to two-dimensions, allowing the incoming subunit to make higher number of bonds with the nucleus. **Lines 266-268, 307-327**

IF so, can it be shown in WT tubulin, by using higher concentrations of free tubulin than 10 μ M?

In theory, raising the tubulin concentration to 60 μ M will allow WT tubulin to nucleate at rate comparable to that of Y222F tubulin at 5 μ M. Unfortunately however, the rapid flush technique did not work at such high concentration because oligomers aggregated. **Lines 266-268**

Response to Reviewer #2

Thank you for reviewing our manuscript entitled “GTP-dependent formation of straight oligomers leads to nucleation of microtubule” (JCB #202007033). The Reviewer’s criticisms helped us refine our model about the structural pathway of nucleation.

Reviewer’s comment: overall view

The authors investigate here the mechanism of microtubule nucleation in solution with purified recombinant Drosophila tubulin. They compare wild type tubulin and a single point mutant that is intended to alter the conformation of a loop at the surface of beta-tubulin in a nucleotide-dependent manner. This loop was previously proposed to mediate longitudinal tubulin interactions. Using turbidity measurements and darkfield microscopy, the authors find that this Y222F mutation does indeed promote microtubule nucleation and growth. They then study tubulin oligomers of wildtype and mutated tubulin in the presence of both GTP and GDP by negative stain electron microscopy and find a negative correlation between oligomer curvature and nucleation efficiency. Comparing oligomer sizes (from EM data) and critical nucleus sizes (from kinetic data) the authors conclude that the critical nucleus required for persistent microtubule growth is formed by the lateral association of 2 straight tubulin oligomers and that oligomer straightening promotes microtubule nucleation by promoting lateral association of oligomers. These are carefully performed experiments, providing new information regarding the long-standing question of the mechanism of microtubule nucleation. However, not all conclusions regarding the mechanism of nucleation appear to be compelling as stated at this stage of the manuscript.

.....In conclusion, the authors present here several very interesting correlations in the context of the nucleation efficiency of different tubulins in different nucleotide states. The model of the authors stating a set of causalities is plausible, but alternative models seem to be also possible. The mechanistic origin of the difference in nucleation efficiency between mutant and wildtype tubulin in the presence of GTP is probably not yet fully understood and could deserve some more discussion.

We agree with the Reviewer’s opinion that the mechanistic origin of the difference in nucleation efficiency between mutant and wild type tubulin in the presence of GTP is not yet fully understood. We revised Results and Discussion, taking all Reviewer’s comments into consideration. We appreciate the Reviewer’s criticism that helped us refine the manuscript. At the same time, we would like to make it clear that the subject of this paper is the mechanism of GTP-dependent nucleation, not the mechanistic origin of the high nucleation efficiency in the Y222F mutant. In revising the manuscript, we tried to keep on the right track, refraining from the excessive discussion about the Y222F mutant.

1) Mechanism by which the Y222F mutation affects nucleation: The Y222F mutation is expected to break an interaction with the T5 loop of beta-tubulin with GDP bound. According to previous data, in wild-type tubulin GTP induces a conformational change naturally breaking the loop interaction, moving the loop 'out' and consequently promoting longitudinal tubulin interactions. Considering this view, the main effect of the mutation should be expected for GDP tubulin. And indeed, the authors find that GDP-Y222F tubulin can polymerize remarkably well - in contrast to wild type tubulin. However, also GTP-Y222F tubulin nucleates and polymerizes better than GTP-wild type tubulin. This does not seem to agree with the view of the role of the T5 loop that the authors present here. For both GTP tubulins, the loop should be 'out' and longitudinal tubulin interactions should be the same/similar for wild type and mutant in the GTP state. It seems unlikely that GTP is hydrolysed at the ends of oligomers (if it was, it should be substantial given the assumed transient nature of oligomers and could be measured). If oligomers do not hydrolyse GTP in significant amounts, then it seems that the mutation does something else to promote tubulin interactions in the GTP state. What is it?

We don’t know if GTP hydrolysis takes place at the ends of oligomers (such a specific hydrolysis would be technically difficult to measure) but it is possible. There are indications that GTP hydrolysis occurs in tubulin oligomers and downregulates MT nucleation (Carlier et al (1997) Biophysical J 73:418-27). Accordingly, tubulin bound to the stable analog GMPCPP (Hyman et al (1992) Mol Biol Cell 3:1155-67) or a tubulin mutant unable to hydrolyze GTP (Roostalu et al (2020) eLife 9:e51992) nucleate MTs more efficiently than WT GTP-tubulin. In the case of Y222F,

one possibility is that MT nucleation is favoured because the effect of GTP hydrolysis is less pronounced than in WT tubulin (T5 does not switch back to the “in” conformation).

We, however, agree with this reviewer about the possibility that the Y222F mutation does something else, but we don't know what it is. To take this reviewer's comment into account, in the new section added to the Discussion (***The mechanism of accelerated nucleation by Y222F mutation***), we discussed two possible scenarios for a mechanism the mutation resulted in long straight oligomers. One possibility is, as mentioned above, that GTP is hydrolyzed at the end of oligomers, causing WT oligomers to curve. The Y222F oligomers remain straight even after GTP hydrolysis because T5 loop does not switch back to the “in” conformation in this mutant. The other possibility is that the local structure around the T5 loop of a tubulin in oligomers could actually be different from the one we observe in the crystal structure. The structure of T5 loop may change with the assembly of Y222F oligomers, allowing higher stability and straightness for larger Y222F oligomers. We do not know the underlying mechanism, but the size distribution of Y222F oligomers suggests a kind of cooperativity in longitudinal inter-dimer interactions (Fig. 5E).

Lines 372-386

2) Effect of oligomer concentrations: The authors focus here on differences in curvature of wild type and Y222F tubulin oligomers and argue that the observed curvature differences cause differences in the nucleation efficiency of different tubulins. If indeed the kinetic pathway to microtubule formation includes the lateral association of oligomers, one expects that the simply concentration of oligomers has a strong effect on microtubule nucleation. The authors observe tubulin oligomers for the Y222F mutant at lower tubulin concentrations than for wild type and in both cases seem to have a convenient density for their morphological analysis by EM. (The authors also estimate oligomer concentrations in Suppl. Fig. 5, but it was unclear if these oligomer concentrations referred to different tubulin concentrations for wild type and mutant.) Does this mean that the Y222F oligomer concentration per tubulin concentration is much higher than for wild type? What are the relative concentrations of oligomers per tubulin concentration for the 4 categories of wild type and Y222F tubulin with each GTP and GDP? Is possibly simply the much higher stability (and therefore concentration) of oligomers the main reason for the increased nucleation efficiency of Y222F tubulin, with the details of minor shape differences between different oligomer classes playing perhaps only a minor role?

The data in the previous Suppl. Fig. 5 (now Fig. 5D-F) represent the distribution of oligomer concentrations, not the relative concentrations of oligomers per tubulin concentrations. What counts for the nucleation rate is the absolute population of oligomers, not the propensity of tubulin to oligomerize (= relative concentration of oligomers per tubulin concentration).

The concentration of Y222F oligomer reaching a critical size is lower than that of WT oligomers reaching a critical size (2.4 nM and 16.5 nM, respectively, Fig. 5D, E), despite the nucleation rate in mutant being three order of magnitude higher than that of WT (2.1×10^{-10} and 2.1×10^{-13} M/s for Y222F and WT, respectively, Fig. 4C), indicating that each single Y222F critical nucleus has higher ability to become a MT than a WT critical nucleus. We added this point in Discussion. **Lines 357-364**

The concentrations of oligomers in Fig. 5D-F have been slightly changed from the previous version to incorporate our most recent calculations. The fitting of the lines in the legend of Fig. 5 and the values of ΔG_{olig} were changed accordingly. **Fig. 5D-F, Lines 534-538, 244-246**

3) Scholarship: In the Introduction, it seems appropriate to clearly introduce previous published observations of tubulin oligomers and previous conclusions of their role for microtubule nucleation and to explain in the Discussion to which extent the observations here agree with previous work and to which extent this study goes beyond previous results and conclusions (e.g. Voter & Erickson, JBC, 1984; Wang et al., Cell Cycle, 2005; Mozziconacci et al., Plos One, 2008; Portran et al., NCB, 2017). In this context, the authors might want to rephrase their sentence in the Introduction that the visualization of nucleation intermediates is "impossible" if not using their method.

Thank you for pointing this out. We revised the Introduction and Discussion, following the reviewer's suggestion.
Lines 114-118, 246-248, 311-313, 331-354

4) *Fig. 2: Why is the structure of wild type tubulin presented with GMPCPP bound and the structure of the Y222F mutant with GTP bound? The authors could be clear about this in the main text. Can the nucleotide difference affect the conclusions drawn for the structure of these tubulins in the GTP state?*

As for the first question, the primary reason is technical. The GMPCPP-WT structure was determined first. The use of GMPCPP makes the crystallization experiments easier since the hydrolysis of this nucleotide is much slower than that of GTP. In the case of the Y222F mutant, we did not succeed in obtaining crystals with bound GMPCPP. Instead, we obtained crystals from GTP-Y222F tubulin (as a T2R complex). We harvested these crystals about 15 hours after setting up the crystallization drops, a compromise between the size of the crystals and the hydrolysis of GTP. This has led to a structure with full GTP occupancy in the nucleotide-binding site of the beta-tubulin at the end of the T2R complex (named "beta2" in Fig. 1C, previously Fig. 2a) and a mix of GDP and GTP (about 50% each) in "beta1" which interacts with the "alpha2" subunit. We inserted "note that we did not succeed in obtaining crystals of Y222F with bound GMPCPP" in the main text (**lines 149-150**) and mentioned about the nucleotide used in crystallization experiments in the Materials and Methods section "Crystallisation and structure determination". **Lines 656-667 in Materials and Methods**

As for the second question, the nucleotide difference is unlikely to affect the conclusions drawn in this study. Specifically, in the case of mammalian brain tubulin (which shares more than 96% sequence identity with *Drosophila* tubulin), both GTP-tubulin and GMPCPP-tubulin structures have been determined and no difference was observed as long as nucleotide hydrolysis remained low (Nawrotek et al (2011) J Mol Biol 412, 35-42). We also want to stress that the main structural difference between WT tubulin and the Y222F mutant is observed with bound GDP, and that in this case the same nucleotide was used.

5) *Fig. 3e-i: Why is the nucleotide-dependent difference between the distributions of oligomer curvature larger for wild type than for Y222F tubulin? How significant at all is the difference between GTP and GDP state for Y222F tubulin?*

As was written in the manuscript, in both WT and Y222F pairs, the statistical significance of the difference between GTP and GDP state is < 0.01 (Mann-Whitney U test; now in **line 194-197**). We do not know if the nucleotide-dependent difference is really larger for WT than the difference for the Y222F mutant, as this reviewer wrote. At the same time, we do not have particular reason to believe that the magnitude of nucleotide-dependent difference should be the same in WT and Y222F mutant.

6) *Same Fig.: Why are oligomers of the mutant in GTP straighter than those of the wild type in GTP?*

This question, related to comment (1), is addressed in the new section of the Discussion (***The mechanism of accelerated nucleation by Y222F mutation***). We discussed two possible scenarios for a mechanism the mutation resulted in long straight oligomers. One possibility is that GTP is hydrolyzed at the end of oligomers, causing WT oligomers to curve, but the Y222F oligomers remain straight because T5 loop does not switch back to the "in" conformation even after GTP hydrolysis. The other possibility is that the structure of the T5 loop may change with the assembly of Y222F oligomers, allowing higher stability and straightness for larger Y222F oligomers. **Lines 372-386**

7) *Calculation of critical nucleus size (starting in line 195): Are oligomers assumed to grow only by plus end elongation? It seems that only plus end growth speeds have been used for the calculation of critical nucleus size. How does the size of the critical nucleus change if one assumes that oligomers can also elongate by minus end growth?*

Because WT(GTP) and Y222F(GTP) MT grow only by the plus end (new Fig. 3E,G, previously Suppl. Fig. 4), we used the growth rate at the plus end as v_0 . Even if the MT growth is bidirectional, it should not affect the estimation of the size of critical nucleus because the linearity of net v_0 (the sum of the growth rates at both ends) as a function of

tubulin concentration is maintained. We do not particularly discuss this point in the manuscript.

Why is the critical nucleus for Y222F tubulin larger than for wild type? In other words, why does the nucleus for the tubulin that nucleates better need to be bigger to be stable? This seems to be counter-intuitive.

This is an interesting point. A larger size of critical nucleus of the Y222F mutant having higher nucleation rate may look paradoxical at first glance. Although we do not have definitive answers with sufficient data, our qualitative understanding is as follows.

Either for WT or Y222F, the critical size is determined by the balance between the entropic cost (to gather tubulin dimers in solution to the site of oligomers) and gain of binding free energy. The nucleation rate is determined by a barrier height at the critical size.

Because the free energy cost of realizing straight oligomer is much larger for WT than that for Y222F, as seen by the less frequent straight conformations of WT, in WT, increasing the oligomer size does not result in lowering its barrier height. While the length favors energy gain owing to lateral binding, it is not large enough to compensate the high entropic cost. Therefore, the WT tubulin has to compromise and find the barrier peak with relatively short oligomers, where the height is higher than that of Y222F tubulin.

With Y222F tubulin, the barrier height can be maintained low (= rapid nucleation) under a larger critical size thanks to the high gain of longitudinal binding energy.

This explanation is included in Discussion. **Lines 302-306, 356-371**

8) *Fig. 5d: Can the oligomer length distributions be related to the oligomer curvature distributions? Are oligomers straighter when they are longer? Taken together, it appears that the potentially nucleation-promoting characteristics of a reduced curvature and a longer length of Y222F oligomers may be compensated by the need for a larger nucleus to form. Maybe instead the larger critical nucleus is a consequence of longer oligomers (since they have to come together laterally to form a nucleus). Considering all these data together, it is a little unclear why Y222F tubulin nucleates better in the model of the authors.*

Can the oligomer length distributions be related to the oligomer curvature distributions? Are oligomers straighter when they are longer?

No, it is not generally the case: Our linear regression analysis indicates that the larger the size of oligomer, the straighter the curvature in Y222F(GTP), but not in other three conditions including WT(GTP).

Taken together, it appears that the potentially nucleation-promoting characteristics of a reduced curvature and a longer length of Y222F oligomers may be compensated by the need for a larger nucleus to form. Maybe instead the larger critical nucleus is a consequence of longer oligomers (since they have to come together laterally to form a

nucleus). Considering all these data together, it is a little unclear why Y222F tubulin nucleates better in the model of the authors.

As our data do not support the reviewer's hypothesis "longer oligomers are straighter", it is impossible to discuss his/her further claim that "Maybe instead the larger critical nucleus is a consequence of longer oligomers". The authors presume that in his/her comment, the reviewer might have wanted to mention that "Maybe instead the straighter critical nucleus is a consequence of longer oligomers" (If it is not the reviewer's original intention, the authors do not understand the paragraph above).

Moreover, if the length, not the curvature, is the major factor affecting the nucleation rate (according to this reviewer, 'nucleation efficiency'), WT(GDP) tubulin should nucleate better than WT(GTP) tubulin (Fig. 3B). Our result clearly shows that WT(GDP) oligomers are longer than WT(GTP) oligomers (Fig. 2F). Our observation is not an artefact; it parallels with the higher stability of GDP-protofilament reported earlier (Valiron et al., 2010, JBC 85:17507). Combining our results in Fig. 1I-P, Fig. 2E-G, Fig. 4 and Fig. 5 together, the simplest interpretation is that the curvature is the major factor controlling the nucleation rate.

To organize the data, together with our interpretation, on Y222F mutant, we added a section entitled, "**The mechanism of accelerated nucleation by Y222F mutation**" in Discussion.

9) *Suppl. Fig. 4: Is the catastrophe frequency for the Y222F mutant reduced, because microtubules grow faster than wild type? A plot of the catastrophe frequency as a function of growth speed for the two tubulins would give a clear answer. Why are Y222F microtubules growing in GTP more stable than wild type microtubule in GTP?*

Is the catastrophe frequency for the Y222F mutant reduced, because microtubules grow faster than wild type?

The lines for WT(GTP) and Y222F(GTP) are (not very far from being) parallel (the one for Y222F(GDP) is on the X-axis, as there is no catastrophe). This means that for a given growth speed, the catastrophe frequency is higher in the case of the WT. Hence Y222F MTs are more stable per se, not because they grow faster.

Why are Y222F microtubules growing in GTP more stable than wild type microtubule in GTP?

This is an important question, but answering this question requires the determination of the Y222F-MT structure, which is beyond the scope of this study.

The following part in Discussion may give some clue to answer the question. "It is possible that any factor that stabilizes straight oligomers/protofilament can accelerate nucleation, and, later in the growing phase, facilitate polymerization and prevent catastrophe at the growing end of MT. In both cases, stable growth of oligomer/protofilament may require the coming tubulin to simultaneously make longitudinal and lateral bonds with an array of straight oligomers/protofilaments." **Lines 425-430**

Response to Reviewer #3

We thank Dr. Zhang for his very positive evaluation of our work. Please find hereunder our response to this reviewer's comments.

1) In the polymerized state, as shown in the high-resolution cryo-EM structures of microtubule, the T5 loop of beta tubulin is in the 'in' conformation, regardless of the nucleotide state at the E-site (Zhang et al. PNAS 2018). This seems to be inconsistent with the major conclusion in this paper - the 'out' conformation of T5 is favored for microtubule nucleation/formation. However, it is highly possible that the T5 flips back "in" once the tubulins are incorporated into the lattice and/or upon GTP hydrolysis. I look forward to the authors' discussion regarding this issue.

Indeed, the beta-tubulin T5 loop is in an “in” conformation when tubulin is embedded in the microtubule core. Actually, such an “in” conformation is required, because an “out” conformation would preclude the establishment of the longitudinal contacts between tubulins along a straight protofilament.

The hypothesis proposed by Dr. Zhang (T5 switching to an “in” conformation upon or soon after incorporation in the lattice) is also the one we favor. In this scenario, the T5 loop of the tubulin molecule at the very end of a protofilament favors the recruitment of a next tubulin (elongation). Once this is done, T5 of this (now) penultimate tubulin does not play a major role anymore and can switch to an “in” conformation. To ascertain this scenario, the T5 loop conformation of tubulin molecules at the end of a protofilament, including that of the very last tubulin, should be established. However, obtaining such data is currently out of reach.

This point is now mentioned in the 2nd paragraph of the discussion and reads: “The GTP-dependent extension of the T5 loop mediates the establishment of the longitudinal interdimer contacts and stabilizes them (Natarajan et al., 2013; Nawrotek et al., 2011), giving rise to a subpopulation of nearly straight oligomers (highlighted in yellow in Fig. 2G). Once these contacts are established, T5 loop may switch back to an “in” conformation, as seen in the microtubule core (Ref Zhang et al, PNAS, 2018). The oligomers also interact laterally...” **Lines 293-297**

2) The authors may consider merging or rearranging Fig. 1 and 2. The current panel Fig. 1a is a bit hard to follow for readers who are unfamiliar with tubulin structure, and it is rare to have a schematic picture as Fig. 1.

We revised the Figure arrangement following the reviewer's suggestion. **Fig.1**

3) The authors wrote 'the negatively charged residue D177 in beta-tubulin is exposed towards the solvent, likely mediating the incoming tubulin dimer having positive charges on the interface (of alpha-tubulin) to establish a longitudinal contact'. Now that high-resolution structures are available for both curved and straight tubulin conformations, can the authors make an attempt to model the tubulin longitudinal interface for the early step of nucleation (i.e. straight oligomers), or at least be more specific about which positively charged residues of alpha tubulin are likely involved? Some of the residues proposed in a previous paper (Nawrotek et al. JMB 2011) seem to be far away from D177 even in a 'out' conformation.

Structural analysis based on the tubulin-tubulin longitudinal interface in the T2R complex indicates that K352 is the alpha-tubulin basic residue closest to beta-tubulin T5 D177 (about 5 Ang distance). K336 would be the closest if T5 were to adopt an “out” conformation at this interface (see Figure for reviewer 3). A structural analysis based on straight protofilaments leads to the identification of the same residues.

These two residues are therefore the most obvious candidates to be involved in this process. They are now explicitly mentioned in the manuscript: “the negatively charged residue D177 in beta-tubulin is exposed towards the solvent, likely mediating the incoming tubulin dimer having positive charges on the alpha-tubulin interface (in particular, from K336 and K352) to establish a longitudinal contact.” **Lines 129-133**

However, additional experiments will be needed to ascertain further the implication of these residues, for instance by recording the effect of their mutation. These experiments are a study on its own if to be carefully performed and therefore go beyond the scope of the present study.

Figure_for reviewer 3. Identification of the alpha-tubulin basic residues closest to beta-tubulin D177. (Left) Overview of the structure of WT(GMPCPP) *Drosophila* tubulin within the T2R complex. This complex comprises two tubulin heterodimers arranged head-to-tail as in a curved protofilament and one stathmin-like protein (magenta). α -tubulin of both heterodimers is in grey, β -tubulin is either in green (β 1 = subunit at the interface with the “second” tubulin of the complex) or in cyan (β 2 = subunit at the end of the complex). (Right) Close-up of the part framed in the left panel. For clarity, helix H8 and strand S8 of α 2 were not traced. K352, belonging to strand S9, is the α 2 basic residue closest to β 1 D177. With the T5 loop modeled in an “out” conformation (in cyan, obtained after superposition of β 2 on β 1), the H10 α -tubulin residue K336 would be close to D177. Note that a T5 “out” conformation would lead to steric conflicts with α 2 unless the relative orientation of the two tubulins of the complex changes.

4) *I am very curious what the 'early nucleation intermediates' of GMPCPP-microtubule look like, as visualized by rapid flush negative stain EM. Perhaps this event is too fast to be captured even by this technique. To be clear, I am not asking for new experiments, given the COVID19 situation, but the authors may consider including the GMPCPP data if they already have them in hand.*

Thank you for your kind thoughts. We had several trials but so far unsuccessful.

5) *The residue number Y222 of beta tubulin is inconsistent with the numbering Y224 in a previous paper by one of the co-authors (Nawrotek et al. JMB 2011). I understand this is due to alignment between alpha and beta tubulin, but it should be noted in the paper.*

Thank you for pointing this out. To deal with the problem, we mentioned that Y222 is equivalent to Y224 in the numbering used in Nawrotek et al. **Lines 110-111**

6) *The current Sup Fig. 1 can be moved to the main figures after figure rearrangement.*

We understand the Reviewer’s intension. However, to keep a seamless story line from Introduction to main Results, we decided not to move the Sup Fig. 1 showing the data belonging to Materials and Methods to main text.

December 10, 2020

RE: JCB Manuscript #202007033R

Dr. Etsuko Muto
RIKEN Center for Brain Science
Hirosawa 2-1
Wako, Saitama 351-0198
Japan

Dear Dr. Muto,

Thank you for submitting your revised manuscript entitled "GTP-dependent formation of straight tubulin oligomers leads to microtubule nucleation". We and the returning reviewer appreciated the changes made in revision, including to address the mechanistic interpretation of the Y222F mutation. We would be happy to publish your paper in JCB pending final revisions necessary to meet our formatting guidelines (see details below) and pending text changes addressing the remaining points of the reviewer.

1) eTOC summary: A 40-word summary that describes the context and significance of the findings for a general readership should be included on the title page. The statement should be written in the present tense and refer to the work in the third person.

Suggested revision to match our preferred style:

" Ayukawa, Iwata, Imai, et al. visualize the early intermediates in the pathway of spontaneous nucleation of microtubules by using rapid flash negative stain electron microscopy. This study demonstrates that the formation of straight tubulin oligomers of critical size is essential for nucleation"

2) Statistical analysis: Error bars on graphic representations of numerical data must be clearly described in the figure legend. The number of independent data points (n) represented in a graph must be indicated in the legend. Statistical methods should be explained in full in the materials and methods. For figures presenting pooled data the statistical measure should be defined in the figure legends.

3) Materials and methods: Should be comprehensive and not simply reference a previous publication for details on how an experiment was performed. Please provide full descriptions in the text for readers who may not have access to referenced manuscripts.

- For all cell lines, vectors, constructs/cDNAs, etc. - all genetic material: please include database / vendor ID (e.g., Addgene, ATCC, etc.) or if unavailable, please briefly describe their basic genetic features *even if described in other published work or gifted to you by other investigators*
- Please include species and source for all antibodies, including secondary, as well as catalog numbers/vendor identifiers if available.
- Sequences should be provided for all oligos: primers, si/shRNA, gRNAs, etc.
- Microscope image acquisition: The following information must be provided about the acquisition

and processing of images:

- a. Make and model of microscope
- b. Type, magnification, and numerical aperture of the objective lenses
- c. Temperature
- d. imaging medium
- e. Fluorochromes
- f. Camera make and model
- g. Acquisition software
- h. Any software used for image processing subsequent to data acquisition. Please include details and types of operations involved (e.g., type of deconvolution, 3D reconstitutions, surface or volume rendering, gamma adjustments, etc.).

A. MANUSCRIPT ORGANIZATION AND FORMATTING:

Full guidelines are available on our Instructions for Authors page, <https://jcb.rupress.org/submission-guidelines#revised>. **Submission of a paper that does not conform to JCB guidelines will delay the acceptance of your manuscript.**

B. FINAL FILES:

-- High-resolution figure and video files: See our detailed guidelines for preparing your production-ready images, <https://jcb.rupress.org/fig-vid-guidelines>.

Thank you for this interesting contribution, we look forward to publishing your paper in Journal of Cell Biology.

Sincerely,

Arshad Desai, PhD
Monitoring Editor, Journal of Cell Biology

Melina Casadio, PhD
Senior Scientific Editor, Journal of Cell Biology

Reviewer #2 (Comments to the Authors (Required)):

The authors have made a very good effort to address the reviewers' concerns. Overall this interesting manuscript has been nicely improved and the presented results advance our understanding of microtubule nucleation.

Two replies to this reviewer's previous major concerns might profit from further clarification:

Concern 2 - effect of oligomer concentration on nucleation rate: The question was: Is it possible that total oligomer concentration (not only the concentration of oligomers above what the authors consider the "critical size") scales with nucleation rate? According to the authors' response, it may not. But it would be useful to state it clearly.

Concern 7 - growth speeds: The authors state that under their experimental conditions only microtubule plus ends grow and support this by showing kymographs of microtubules growing at a particular concentration, arguing that therefore considering only plus end growth is sufficient for their analysis. Is microtubule minus end growth absent over the entire concentration range studied? If not, how does it affect the estimation of critical nucleus size (particularly of wt microtubules that might show minus end growth at higher concentrations)?

Three remaining minor concerns referring to statements in the Introduction:

"...solved a long-standing important question: how does GTP-tubulin nucleate MTs in vitro?" Does a statement in this generality appropriately capture the advance this study makes here or would a more specific statement be clearer?

"...tacit assumption held by cell biologists that the mechanism of spontaneous nucleation in vitro has little relevance to...". Some cell biologists might feel misinterpreted here, particularly some of those whose publications are cited to support the statement; a more carefully worded statement could reflect the current thinking in the field more accurately.

"their visualization is impossible by ordinary imaging technique". This statement probably does not intend to imply that oligomers could not be observed in the past, given that the authors cite such studies later in the Introduction. More precise wording might help to clarify what exactly has been impossible previously.

Response to Reviewer #2

We thank this reviewer for her/his very positive evaluation of our work. Please find hereunder our response to this reviewer's comments.

Reviewer's comment

Two replies to this reviewer's previous major concerns might profit from further clarification:

Concern 2 - effect of oligomer concentration on nucleation rate: The question was: Is it possible that total oligomer concentration (not only the concentration of oligomers above what the authors consider the "critical size") scales with nucleation rate? According to the authors' response, it may not. But it would be useful to state it clearly.

We revised the statement in Discussion accordingly (the first sentence in the section "*The mechanism of accelerated nucleation by Y222F mutation*" in Discussion).

Concern 7 - growth speeds: The authors state that under their experimental conditions only microtubule plus ends grow and support this by showing kymographs of microtubules growing at a particular concentration, arguing that therefore considering only plus end growth is sufficient for their analysis. Is microtubule minus end growth absent over the entire concentration range studied? If not, how does it affect the estimation of critical nucleus size (particularly of wt microtubules that might show minus end growth at higher concentrations)?

The minus end growth was absent over the entire concentration range studied.

Three remaining minor concerns referring to statements in the Introduction:

"...solved a long-standing important question: how does GTP-tubulin nucleate MTs in vitro?" Does a statement in this generality appropriately capture the advance this study makes here or would a more specific statement be clearer?

Thank you for the suggestion. Because the purpose of Introduction is to interest broad range of readers, we prefer to keep the current simple statement.

"...tacit assumption held by cell biologists that the mechanism of spontaneous nucleation in vitro has little relevance to...". Some cell biologists might feel misinterpreted here, particularly some of those whose publications are cited to support the statement; a more carefully worded statement could reflect the current thinking in the field more accurately.

We agree that the cited references and the text were not properly coordinated. We revised the paragraph spanning p.4 and p.5

"their visualization is impossible by ordinary imaging technique". This statement probably does not intend to imply that oligomers could not be observed in the past, given that the authors cite such studies later in the Introduction. More precise wording might help to clarify what exactly has been impossible previously.

The text was revised accordingly.